# Gradient Preconditioning for Efficient and Reliable Reward-Guided Generation

**Jisung Hwang** [1]   **Minhyuk Sung** [1]

## Abstract

We propose a gradient preconditioning method that makes reward-guided generation with one-step generative models both efficient and reliable. Test-time noise optimization can unlock substantially better reward-guided generations from pretrained generative models, but it is prone to reward hacking that degrades quality and is often too slow for practical use. We precondition reward gradients by projecting them onto a carefully designed white Gaussian noise feasible set, a compact spectral set with blockwise norm constraints that tightly captures the statistics and spatial uncorrelatedness of white Gaussian noise. This preconditioning reshapes each gradient update into a noise-aligned direction, driving faster and more effective reward ascent while preventing reward hacking. The projection is closed-form and matches the $\mathcal{O}(N \log N)$ complexity of FFT, adding negligible overhead in practice. In experiments on FLUX with four reward models, our approach reaches a comparable Aesthetic Score using only 30% of the wall-clock time required by the state-of-the-art regularization-based method.

## 1. Introduction

Deep generative models for high-dimensional continuous data typically rely on a simple but powerful design choice: sampling a latent vector from the *standard Gaussian distribution*. In practice, this "white Gaussian noise" prior is not only a mathematical convenience but also a key ingredient that supports stable and realistic generation. Recently, many applications have moved beyond pure generation to reward-guided sampling with a task-specific reward (Min et al., 2025; Ben-Hamu et al., 2024; Tang et al., 2024). In particular, distillation techniques for diffusion and flow models, such as shortcut and consistency models (Labs, 2024; Sauer et al., 2024; Frans et al., 2025), enable *one-step* generation,

which makes it easy to directly optimize the latent vector at test time (Eyring et al., 2024; Hwang et al., 2025; Kim et al., 2025) to maximize the reward. This *test-time latent optimization* is appealing because it can effectively exploit the capacity of a pretrained model without any retraining, while leveraging the fast generation speed of a one-step model for test-time search.

Despite its promise, test-time optimization with one-step generative models has two major practical limitations. First, it is *unreliable*: as optimization progresses, the latent vector can drift away from white Gaussian noise, and the model may produce unrealistic artifacts, a phenomenon commonly referred to as *reward hacking*. Second, it is *slow*: existing methods typically require multiple gradient steps per sample. Thus, even with one-step models, producing a single output can take minutes. Making test-time optimization both efficient and reliable is therefore crucial for practical reward-guided generation.

Existing approaches attempt to address these issues by adding *regularization* terms that encourage Gaussian-related properties during optimization (Eyring et al., 2024; Tang et al., 2024; Hwang et al., 2025). While effective in many cases, regularization is inherently a *soft* mechanism. It provides no guarantee that the latent vector remains noise-like, and it introduces additional weights that must be tuned. Moreover, when the optimizer discovers shortcuts, soft penalties can be insufficient to prevent reward hacking.

We propose a simple and practical alternative: **gradient preconditioning** via a carefully designed white Gaussian noise feasible set. We construct a compact spectral feasible set that tightly characterizes white Gaussian noise, enforcing blockwise norm constraints on DFT coefficients to capture both the marginal Gaussian statistics and the spatial uncorrelatedness of white noise. At each optimization step, the reward gradient is projected onto this feasible set, reshaping each update into a noise-aligned direction. This preconditioning drives faster and more effective reward ascent while preventing reward hacking, without introducing any additional hyperparameters. Crucially, the projection is closed-form with $\mathcal{O}(N \log N)$ complexity, comparable to widely used algorithms such as sorting or FFT, and in practice accounts for only 0.04% of the runtime in our experiments with the FLUX model.

---

[1]KAIST, Daejeon, Republic of Korea. Correspondence to: Minhyuk Sung <mhsung@kaist.ac.kr>.

*Proceedings of the 43rd International Conference on Machine Learning*, Seoul, South Korea. PMLR 306, 2026. Copyright 2026 by the author(s).

Empirically, we validate our method on test-time latent optimization for one-step text-to-image models, following the standard setup used in prior work such as ReNO (Eyring et al., 2024) and MPGR (Hwang et al., 2025). Across multiple reward models, our gradient preconditioning achieves higher target rewards while preserving held-out human-preference metrics (Aesthetic Score, PickScore, HPSv2, ImageReward), yielding stable and realistic samples without reward hacking. Notably, on FLUX (Labs, 2024) for Aesthetic Score optimization, our approach reaches a comparable reward level using only **30% of the wall-clock time** required by the state-of-the-art regularization-based method MPGR. By resolving both the efficiency bottleneck and reward hacking, we make test-time latent optimization substantially more practical.

## 2. Related Work

Reward-guided generation is commonly approached in two complementary ways. One direction *fine-tunes* or *trains* a model (Black et al., 2024; Clark et al., 2024; Wallace et al., 2024) for a specific reward, but this typically requires substantial compute, from tens (Eyring et al., 2025) to hundreds of GPU hours (Liu et al., 2025a), and must be repeated per reward. The other direction performs *test-time optimization* (Karunratanakul et al., 2024; Zhao et al., 2025; Min et al., 2025; Kim et al., 2025) without retraining, which is lighter-weight and can be applied to different rewards at no additional training cost. In practice, these two directions are orthogonal; a fine-tuned model can still benefit from test-time optimization for further improvement (Liu et al., 2025b).

The practicality of test-time optimization depends on the generative model. For *multi-step* diffusion or flow models, reward evaluation at intermediate states is either unreliable when relying on Tweedie's estimate or very expensive when performing all the denoising steps to obtain a clean sample (Ben-Hamu et al., 2024; Oshima et al., 2025). In contrast, *one-step* generative models enable efficient latent optimization since each reward evaluation can be done with a single function evaluation and uses a cleaner output (Eyring et al., 2024; Hwang et al., 2025). We therefore focus on one-step models in our experiments.

In one-step models, latent optimization can be performed directly in the noise prior, which offers the widest search space. Preserving white Gaussian noise characteristics is therefore crucial for stable and realistic generation. In the following, we review prior regularization methods that motivate our white Gaussian noise feasible set design.

**Regularizing Marginal Distribution.** Regularization strategies in machine learning are often designed to enforce statistical properties on model weights during optimization, such as through KL divergence (Kingma & Welling, 2014) or kurtosis (Shkolnik et al., 2020), but these regularization terms can also be incorporated into test-time latent optimization. For reward-guided generation, prior works (Samuel et al., 2023; Ben-Hamu et al., 2024; Eyring et al., 2024) regularize the $\ell_2$ norm of the latent vector to encourage alignment with the statistics of the Gaussian distribution. However, these marginal-based approaches treat the latent as an unordered set and therefore fail to regularize *spatial correlation*, which is essential for preserving the property of white Gaussian noise.

**Regularizing Both Marginal and Spatial Structure.** Structure-aware methods regularize both the marginal distribution and spatial correlation by treating the latent vector as an ordered sequence. For example, PRNO (Tang et al., 2024) penalizes the mean and covariance of latent subvectors, promoting spatial decorrelation and Gaussian statistics. Building on the spectral characterization of white noise via flat power spectra (Khintchine, 1934; Oppenheim et al., 1999; Stoica & Moses, 2005), MPGR (Hwang et al., 2025) proposes a spectral regularizer that penalizes deviations in blockwise $\ell_1$ norms of Fourier coefficients from Gaussian statistics. This spectral domain regularization more effectively reduces spatial correlations and preserves white Gaussian noise characteristics. We therefore characterize white Gaussian noise in the spectral domain. In contrast to these regularization-based approaches, we design a white Gaussian noise feasible set that enables efficient gradient preconditioning.

## 3. Gradient Preconditioning for Reward-Guided Generation

Given a generative model $\mathcal{M} : \mathbb{R}^N \to \mathbb{F}$ that maps a latent vector to the data space $\mathbb{F}$, and a reward function $r : \mathbb{F} \to \mathbb{R}$ that scores generated outputs, the goal is to find $\boldsymbol{x} \in \mathbb{R}^N$ that maximizes $r(\mathcal{M}(\boldsymbol{x}))$. Starting from white Gaussian noise, unconstrained gradient ascent causes the latent to drift away from noise-like characteristics, leading to reward hacking and unrealistic outputs. To prevent this, a common approach adds a regularization term $\mathcal{L}_{\text{reg}}$ that encourages $\boldsymbol{x}$ to retain certain properties of white Gaussian noise:

$$\max_{\boldsymbol{x} \in \mathbb{R}^N} \; r(\mathcal{M}(\boldsymbol{x})) - \lambda \mathcal{L}_{\text{reg}}(\boldsymbol{x}), \qquad (1)$$

where $\lambda$ is a hyperparameter. For example, $\mathcal{L}_{\text{reg}}$ may penalize $\ell_2$ norm deviations (Samuel et al., 2023), or blockwise $\ell_1$ norm deviations in the spectral domain (Hwang et al., 2025), relative to Gaussian statistics.

In practice, the optimization is performed via gradient ascent due to the complexity of the generative models. However, it is unclear how effectively $\mathcal{L}_{\text{reg}}$ is minimized during optimization. As a result, regularization may fail to enforce the

---

**Algorithm 1** Gradient-Preconditioned Optimization

    **Iterate:**
        $J \leftarrow r(\mathcal{M}(\boldsymbol{x}))$
        $\boldsymbol{x} \leftarrow \text{Adam}\big(\boldsymbol{x}, \text{Proj}_{\mathcal{G}}(\nabla_{\boldsymbol{x}} J)\big)$

---

characteristics of white Gaussian noise, or conversely, may hinder reward maximization.

We propose a different approach: rather than penalizing deviations from white Gaussian noise, we **precondition the reward gradient** by projecting it onto a white Gaussian noise feasible set $\mathcal{G} \subset \mathbb{R}^N$ before applying the update (Algorithm 1).

At each step, the reward gradient $\nabla_{\boldsymbol{x}} J$ is projected onto $\mathcal{G}$, and the result is used as the update direction. This differs from both regularization and constrained optimization: no additional hyperparameter $\lambda$ is needed, and the latent $\boldsymbol{x}$ itself is not required to lie in $\mathcal{G}$. By steering each update into a noise-aligned direction, gradient preconditioning prevents the optimization from drifting toward reward-hacked solutions, while focusing updates on directions that increase the reward within the noise-compatible subspace to enable faster and more effective reward ascent.

The effectiveness of this approach depends critically on the design of $\mathcal{G}$. Two properties are essential: **(i) efficient closed-form projection**, since the projection is applied at every iteration and must add negligible overhead, and **(ii) tight characterization of white Gaussian noise**, so that projected gradients are genuinely noise-aligned. Simply defining $\mathcal{G}$ as the minimizer set of an existing regularization term $\mathcal{L}_{\text{reg}}$ is generally insufficient, as such sets rarely admit closed-form projections. In the following section, we design a white Gaussian noise feasible set that satisfies both criteria.

## 4. White Gaussian Noise Constraints

In this section, we define and analyze constraints that tightly characterize white Gaussian noise while still admitting an efficient closed-form projection. We build on a prior work that regularizes in the spectral domain and extend it into a gradient preconditioning framework with a closed-form projection operator.

**Notation.** Let $N = 2PB$ be the dimension of a latent vector $\boldsymbol{x} \in \mathbb{R}^N$, where $B$ denotes the block size. The $p$-th block subvector is defined as $\boldsymbol{x}^{(p)} = [x_{pB}, \ldots, x_{pB+B-1}]^\top$. Let $\boldsymbol{F} \in \mathbb{C}^{N \times N}$ be the unitary Discrete Fourier Transform (DFT) matrix, and define the DFT coefficients by $\hat{\boldsymbol{x}} = \boldsymbol{F}\boldsymbol{x}$, which can be efficiently computed using the Fast Fourier Transform (FFT) in $\mathcal{O}(N \log N)$ time.

Our starting point is the regularization term proposed by MPGR (Hwang et al., 2025), which operates in the spectral

domain and encourages each block of DFT coefficients to match the statistics of a complex Gaussian distribution. Concretely, they introduce a blockwise $\ell_1$ penalty

$$\mathcal{L}_{\text{power}}(\boldsymbol{x}) = \frac{1}{N} \sum_{p=0}^{2P-1} \left| \left\| \hat{\boldsymbol{x}}^{(p)} \right\|_1 - \mu B \right|, \qquad (2)$$

where $\mu = 0.875$ (close to $\frac{\sqrt{\pi}}{2} \approx 0.886$). This design is based on two observations: (i) for Gaussian latent vectors, not only the marginal distribution but also the spatial correlation is important, and (ii) these correlations are handled in the spectral domain, where they appear in the DFT magnitudes. By regularizing blockwise statistics of $\hat{\boldsymbol{x}}$, $\mathcal{L}_{\text{power}}$ captures part of this structure.

However, directly turning $\mathcal{L}_{\text{power}}$ into a hard constraint, i.e., enforcing $\mathcal{L}_{\text{power}}(\boldsymbol{x}) = 0$, leads to a feasible set for which no efficient closed-form projection is known. The main difficulty stems from the Hermitian symmetry of the DFT coefficients: different blocks of $\hat{\boldsymbol{x}}$ are correlated, and some coefficients are real-valued while others are complex-valued. This symmetry makes it challenging to derive an efficient closed-form projection algorithm.

To overcome this obstacle, we first define a *compact spectral domain* that removes the Hermitian redundancy while preserving all independent degrees of freedom (Section 4.1). This is achieved via a bijective mapping that aggregates the independent DFT coefficients into a complex-valued vector.

On this compact spectral domain, we then define blockwise constraints that follow the spirit of $\mathcal{L}_{\text{power}}$ but are formulated directly in terms of Gaussian statistics (Section 4.2). Using the relation between the spatial and compact spectral domains, we derive an efficient projection algorithm onto the feasible set (Section 4.3). We then relate our constraints to existing regularization-based approaches and show that they provide a tighter characterization of white Gaussian noise (Section 4.4). Finally, we offer complementary perspectives that connect our constraints to the classical notion of white noise and to the reduction of autocorrelation (Section 4.5).

We refer to our constraints as the **white Gaussian noise constraints**, as they enforce the properties of the Gaussian distribution while ensuring that no frequency component dominates, consistent with the notion of white noise.

### 4.1. Bijective Mapping to Compact Spectral Domain

Our first step is to remove the Hermitian redundancy that arises in the DFT of real-valued latent vectors. This redundancy couples different frequency components and complicates the derivation of a closed-form projection. To address this, we construct a compact spectral domain that contains only the independent DFT coefficients while preserving all information in the spatial domain.

Recall that $\hat{\boldsymbol{x}} = \boldsymbol{F}\boldsymbol{x}$ denotes the DFT of $\boldsymbol{x} \in \mathbb{R}^N$ with even $N$. Then, the DFT coefficients $\hat{\boldsymbol{x}}$ satisfy Hermitian symmetry (Lemma A.1):

$$\hat{x}_0, \, \hat{x}_{N/2} \in \mathbb{R} \quad \text{and} \quad \hat{x}_k = \overline{\hat{x}_{N-k}} \qquad (3)$$

for $k = 1, \ldots, N/2 - 1$. Thus, only $N/2$ complex degrees of freedom are independent: $\hat{x}_0$ and $\hat{x}_{N/2}$, and the first half of the nontrivial frequency coefficients $\hat{x}_1, \ldots, \hat{x}_{N/2-1}$. The remaining entries are fully determined by conjugate symmetry. This redundancy complicates the direct formulation of constraints on $\hat{\boldsymbol{x}}$, particularly when deriving closed-form projections, since one must handle these dependencies.

To resolve this, we define the mapping $\mathcal{F} : \mathbb{R}^N \to \mathbb{C}^{N/2}$ as:

$$\boldsymbol{y} = \mathcal{F}(\boldsymbol{x}) \quad \Longleftrightarrow \quad y_0 = \frac{\hat{x}_0}{\sqrt{2}} + \frac{\hat{x}_{N/2}}{\sqrt{2}} i, \quad y_k = \hat{x}_k \quad (4)$$

for $k = 1, \ldots, N/2 - 1$. This mapping eliminates the redundancy in $\hat{\boldsymbol{x}}$ while preserving all independent components. Specifically, it selects the first half of the complex-valued DFT coefficients and combines the two real-valued terms, $\hat{x}_0$ and $\hat{x}_{N/2}$, into a single complex number. Intuitively, $\mathcal{F}$ yields a compact spectral domain representation with no internal redundancy.

Importantly, this mapping satisfies the following properties:

**Proposition 4.1.** *The mapping $\mathcal{F}$ is a bijection from $\mathbb{R}^N$ to $\mathbb{C}^{N/2}$. Moreover, if $\boldsymbol{z} \sim \mathcal{CN}(\boldsymbol{0}, \boldsymbol{I}_{N/2})$, then $\mathcal{F}^{-1}(\boldsymbol{z}) \sim \mathcal{N}(\boldsymbol{0}, \boldsymbol{I}_N)$.*

*Proof.* See Appendix B. $\qquad\qquad\qquad\qquad\square$

By Proposition 4.1, enforcing that $\boldsymbol{y} = \mathcal{F}(\boldsymbol{x})$ follows the distribution $\mathcal{CN}(\boldsymbol{0}, \boldsymbol{I}_{N/2})$ is equivalent to enforcing that $\boldsymbol{x}$ follows $\mathcal{N}(\boldsymbol{0}, \boldsymbol{I}_N)$. Leveraging this property, we define the spectral domain constraints in the next subsection.

### 4.2. Modeling White Gaussian Noise Constraints

On the compact spectral domain defined by $\mathcal{F}$, we now specify the constraints that define our feasible set. The design follows the *blockwise* spectral regularization $\mathcal{L}_{\text{power}}$ (Equation 2): instead of acting on the entire spectrum at once, we partition the DFT coefficients into local blocks of size $B$ and constrain their aggregate statistics.

Compared to imposing only *global* constraints (e.g., matching the total $\ell_1$ and $\ell_2$ norms of the entire vector), the blockwise constraints define a strictly smaller feasible set. Global constraints fix only two scalar quantities for all coefficients combined, so many different configurations remain admissible. In contrast, our formulation fixes the $\ell_1$ and $\ell_2$ norms *in every block*, introducing $2P$ equality constraints instead of just two. As a result, the feasible set under blockwise constraints is a proper subset of the set defined by the corresponding global constraints, and therefore provides a tighter characterization.

Assume $\boldsymbol{y} \sim \mathcal{CN}(\boldsymbol{0}, \boldsymbol{I}_{N/2})$, which is equivalent to each element $y_j$ being an i.i.d. sample from $\mathcal{CN}(0, 1)$. Then any block $\boldsymbol{y}^{(p)}$ of size $B$ has the expected blockwise norms

$$\mathbb{E}\left[\|\boldsymbol{y}^{(p)}\|_1\right] = \frac{\sqrt{\pi}}{2} B, \qquad \mathbb{E}\left[\|\boldsymbol{y}^{(p)}\|_2^2\right] = B. \quad (5)$$

Using these theoretical statistics of $\mathcal{CN}(0, 1)$, we impose **blockwise $\ell_1$ and $\ell_2$ norm constraints in the compact spectral domain**, requiring that within every block the norms exactly match their theoretical expectations under $\mathcal{CN}(0, 1)$. We use the same block size $B = 16$ as (Hwang et al., 2025). These constraints define the compact spectral domain feasible set

$$\mathcal{G}_{\mathbb{C}} = \Big\{ \boldsymbol{y} \in \mathbb{C}^{N/2} : \|\boldsymbol{y}^{(p)}\|_1 = \frac{\sqrt{\pi}}{2} B, \quad \|\boldsymbol{y}^{(p)}\|_2^2 = B,$$
$$p = 0, \ldots, P - 1 \Big\}. \quad (6)$$

Finally, we lift the definition back to the spatial domain.

$$\boxed{\mathcal{G}_{\mathbb{R}} = \mathcal{F}^{-1}(\mathcal{G}_{\mathbb{C}})} \qquad (7)$$

We refer to the blockwise $\ell_1$ and $\ell_2$ norm constraints in the compact spectral domain as the **white Gaussian noise constraints**. They not only follow the Gaussian statistics but also prevent any single frequency component from dominating in magnitude, thereby aligning with the notion of white noise. Correspondingly, we call the spatial feasible set $\mathcal{G}_{\mathbb{R}}$ the **white Gaussian noise feasible set**. A more detailed interpretation from the perspective of white noise is provided in Section 4.5.

### 4.3. Projection onto the White Gaussian Noise Feasible Set

In this section, we introduce the closed-form projection onto the feasible set $\mathcal{G}_{\mathbb{R}}$ (Equation 7). Specifically, given an input $\boldsymbol{x} \in \mathbb{R}^N$, we aim to find its projection $\dot{\boldsymbol{x}} \in \mathcal{G}_{\mathbb{R}}$ such that:

$$\dot{\boldsymbol{x}} = \operatorname*{argmin}_{\tilde{\boldsymbol{x}} \in \mathbb{R}^N} \|\boldsymbol{x} - \tilde{\boldsymbol{x}}\|_2^2 \quad \text{subject to} \quad \tilde{\boldsymbol{x}} \in \mathcal{G}_{\mathbb{R}}. \quad (8)$$

Since $\mathcal{G}_{\mathbb{R}}$ is defined via spectral constraints, the projection is naturally formulated in the compact spectral domain. Letting $\boldsymbol{y} = \mathcal{F}(\boldsymbol{x})$ and $\dot{\boldsymbol{y}} = \mathcal{F}(\dot{\boldsymbol{x}})$, we obtain:

$$\dot{\boldsymbol{y}} = \operatorname*{argmin}_{\tilde{\boldsymbol{y}} \in \mathbb{C}^{N/2}} \|\mathcal{F}^{-1}(\boldsymbol{y}) - \mathcal{F}^{-1}(\tilde{\boldsymbol{y}})\|_2^2 \quad \text{subject to} \quad \tilde{\boldsymbol{y}} \in \mathcal{G}_{\mathbb{C}}. \quad (9)$$

To simplify this problem, we can utilize the following property of the mapping $\mathcal{F}^{-1}$:

**Proposition 4.2.** *The mapping $\mathcal{F}^{-1}$ is $\mathbb{R}$-linear, and for any $\boldsymbol{z} \in \mathbb{C}^{N/2}$, $\|\mathcal{F}^{-1}(\boldsymbol{z})\|_2^2 = 2\|\boldsymbol{z}\|_2^2$.*

*Proof.* See Appendix C. $\qquad\qquad\qquad\qquad\square$

By Proposition 4.2, we obtain $\|\mathcal{F}^{-1}(\boldsymbol{y}) - \mathcal{F}^{-1}(\tilde{\boldsymbol{y}})\|_2^2 = \|\mathcal{F}^{-1}(\boldsymbol{y} - \tilde{\boldsymbol{y}})\|_2^2 = 2\|\boldsymbol{y} - \tilde{\boldsymbol{y}}\|_2^2$.

Thus, the original projection problem reduces to:

$$\dot{\boldsymbol{y}} = \underset{\tilde{\boldsymbol{y}} \in \mathbb{C}^{N/2}}{\operatorname{argmin}} \|\boldsymbol{y} - \tilde{\boldsymbol{y}}\|_2^2 \quad \text{subject to} \quad \tilde{\boldsymbol{y}} \in \mathcal{G}_{\mathbb{C}}. \quad (10)$$

This corresponds to projecting onto the set $\mathcal{G}_{\mathbb{C}}$, and the projection can be performed independently on each block. In other words, the projection onto $\mathcal{G}_{\mathbb{R}}$ in the spatial domain reduces to a projection onto $\mathcal{G}_{\mathbb{C}}$ in the compact spectral domain, which further decomposes into blockwise projections onto the intersection of $\ell_1$ and $\ell_2$ spheres. The closed-form projection onto this intersection is known to exist (Liu et al., 2020). We derive the projection algorithm for $\mathcal{G}_{\mathbb{C}}$ in Appendix D. Here, we summarize the resulting algorithm.

For the $p$-th block, $\boldsymbol{y}^{(p)}$, consisting of the values from $y_{pB}$ to $y_{pB+B-1}$, we define $\boldsymbol{w}$ as the descending-sorted array

$$\boldsymbol{w} = \texttt{sort\_descending}(\{|y_{pB}|,\ldots,|y_{pB+B-1}|\}) \quad (11)$$

and the cumulative sums $S_{d,k} = \sum_{l=0}^{k} w_l^d$ for $d = 1, 2$.

Then, for $k+1 \geq \frac{\pi}{4}B$, there exists a unique $k^*$ satisfying the following condition (define $w_B = -\infty$):

$$w_{k^*+1} \leq \lambda^{(k^*)} < w_{k^*}, \quad (12)$$

$$\lambda^{(k^*)} = \frac{S_{1,k^*}}{k^*+1} - \frac{\sqrt{\pi}}{2}\frac{\sqrt{B}}{k^*+1}\sqrt{\frac{(k^*+1)S_{2,k^*} - S_{1,k^*}^2}{k^*+1-\frac{\pi}{4}B}}. \quad (13)$$

The final solution is then given by:

$$\dot{y}_j = \frac{\sqrt{\pi}}{2}B\frac{\text{ReLU}\left(|y_j| - \lambda^{(k^*)}\right)}{S_{1,k^*} - (k^*+1)\lambda^{(k^*)}}\frac{y_j}{|y_j|} \quad (14)$$

for $j = pB, \ldots, pB + B - 1$. The projection onto $\mathcal{G}_{\mathbb{C}}$ has computational complexity $\mathcal{O}(N \log B)$. See Appendix D.1 for a detailed explanation.

To summarize, the projection onto $\mathcal{G}_{\mathbb{R}}$ involves computing $\boldsymbol{y} = \mathcal{F}(\boldsymbol{x})$, projecting onto $\mathcal{G}_{\mathbb{C}}$ to obtain $\dot{\boldsymbol{y}}$, and recovering $\dot{\boldsymbol{x}} = \mathcal{F}^{-1}(\dot{\boldsymbol{y}})$. Since $\mathcal{F}$ and its inverse are implemented via the Fast Fourier Transform (FFT), the total procedure takes $\mathcal{O}(N \log N)$ time, which accounts for only **0.04% of the runtime** in our experiments with the FLUX model.

Using this projection, we generated 1M samples of $\boldsymbol{x} \sim \mathcal{N}(\boldsymbol{0}, \boldsymbol{I}_N)$ with $N = 65{,}536$, and computed the cosine similarity between $\boldsymbol{x}$ and its projection onto $\mathcal{G}_{\mathbb{R}}$. The results showed a **minimum cosine similarity of 0.988**, indicating that white Gaussian noise is close to the feasible set $\mathcal{G}_{\mathbb{R}}$.

### 4.4. Connections to Prior Regularization Methods

We explain the connection between our constraints and prior regularization terms, including $\ell_2$ norm regularization and spectral-domain regularization. We then show that our blockwise $\ell_2$ and $\ell_1$ norm constraints are analogous to minimizing these objectives, respectively. This reveals that our constraints provide a tighter characterization of white Gaussian noise and enable efficient closed-form projection.

$\ell_2$ **Norm Regularization (Samuel et al., 2023).** $\ell_2$ norm regularization aims to maximize the likelihood of the Euclidean norm $\|\boldsymbol{x}\|_2$ under the assumption that $\boldsymbol{x} \sim \mathcal{N}(\boldsymbol{0}, \boldsymbol{I}_N)$, in which case $\|\boldsymbol{x}\|_2 \sim \chi_N$, by using the loss:

$$\mathcal{L}_{\text{norm}}(\boldsymbol{x}) = -\log p_{\chi_N}(\|\boldsymbol{x}\|_2), \quad (15)$$

$$p_{\chi_N}(r) = \frac{1}{2^{N/2-1}\Gamma(N/2)} r^{N-1}e^{-r^2/2}, \quad r \geq 0. \quad (16)$$

This loss is minimized when $\|\boldsymbol{x}\|_2^2 = N - 1$, which corresponds to vectors lying on a hypersphere. For high-dimensional generative models such as FLUX ($N = 65{,}536$) and SDXL-Turbo ($N = 16{,}384$), this is nearly equal to the expectation $\mathbb{E}[\|\boldsymbol{x}\|_2^2] = N$.

Our $\ell_2$ norm constraints, $\|\boldsymbol{y}^{(p)}\|_2 = \sqrt{B}$, closely approximate the minimum of $\mathcal{L}_{\text{norm}}$, given that

$$\|\boldsymbol{x}\|_2^2 = \|\mathcal{F}^{-1}(\boldsymbol{y})\|_2^2 = 2\|\boldsymbol{y}\|_2^2 = 2\sum_{p=0}^{P-1}\|\boldsymbol{y}^{(p)}\|_2^2 \quad (17)$$

by Proposition 4.2. Thus, $\boldsymbol{x} \in \mathcal{G}_{\mathbb{R}}$ ensures $\|\boldsymbol{x}\|_2^2 = 2PB = N$ which differs only slightly from the optimal value $N - 1$. Although our constraints are imposed in the spectral domain, they inherently influence the spatial domain as well.

The key distinction is that $\ell_2$ norm regularization, while easily minimized via scaling, does not constrain spatial correlations (Figure 1). In contrast, our formulation enforces local spectral structure, resulting in a tighter characterization of white Gaussian noise.

**Spectral Domain Regularization (Hwang et al., 2025).** The spectral domain regularization term, $\mathcal{L}_{\text{power}}$ (Equation 2), penalizes deviations in the blockwise $\ell_1$ norms of the DFT coefficients.

This regularization exploits the statistical properties of the complex Gaussian distribution $\mathcal{CN}(\boldsymbol{0}, \boldsymbol{I}_B)$ and is similar to our approach in its use of blockwise $\ell_1$ norms. However, our formulation further incorporates blockwise $\ell_2$ constraints, providing a tighter characterization of white Gaussian noise.

The key distinction between (Hwang et al., 2025) and our approach lies in the spectral representation. While $\mathcal{L}_{\text{power}}$ directly regularizes the DFT coefficients $\hat{\boldsymbol{x}} = \boldsymbol{F}\boldsymbol{x}$, we instead use the compact representation $\boldsymbol{y} = \mathcal{F}(\boldsymbol{x})$. This choice allows us to directly define a tractable feasible set $\mathcal{G}_{\mathbb{C}}$ and derive an efficient projection even with a tighter characterization of white Gaussian noise. As a result, projection onto the minimizer set of $\mathcal{L}_{\text{power}}$ requires running gradient descent for hundreds of iterations and therefore is significantly

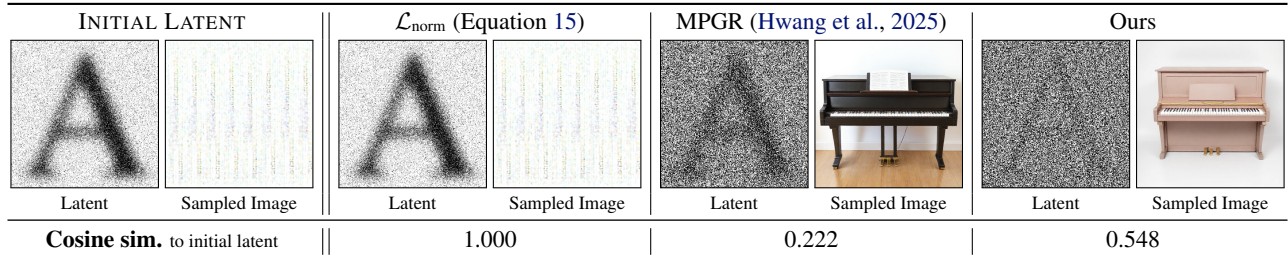

| INITIAL LATENT | | $\mathcal{L}_{\text{norm}}$ (Equation 15) | | MPGR (Hwang et al., 2025) | | Ours | |
|---|---|---|---|---|---|---|---|
| Latent | Sampled Image | Latent | Sampled Image | Latent | Sampled Image | Latent | Sampled Image |
| **Cosine sim.** to initial latent | | 1.000 | | 0.222 | | 0.548 | |

*Figure 1.* **Effectiveness of Projection onto $\mathcal{G}_{\mathbb{R}}$.** Starting from an initial latent encoding the letter 'A', we compare two regularization methods and our projection. Our method preserves high cosine similarity to the initial latent while reducing the spatial correlations. Unlike (Hwang et al., 2025), which requires slow gradient-based iterative projection, our method guarantees optimality with a single operation. The images are sampled from FLUX with the prompt "Piano".

slower. In contrast, our formulation achieves projection with a single lightweight operation, avoiding heavy and imprecise additional computations, as shown in Figure 1.

### 4.5. Interpretation from Multiple Perspectives

Although we have so far explained our constraints using properties of the Gaussian distribution, they can also be interpreted from several other perspectives. We summarize the interpretations below.

**White Noise Perspective.** The constraints prevent any single DFT coefficient from becoming dominant at a specific frequency by limiting the total budget within each block. For example, when $B = 16$, the theoretical maximum of $|y_j|^2$ is approximately 7.18 (see Appendix E), whereas the total budget is $N/2$ for typical $N \gg 10^4$. This ensures that no individual frequency can disproportionately dominate the spectrum. As a result, the constraints prevent the signal from exhibiting strong localized patterns. This mimics the characteristics of white noise, where all samples are independently drawn and no strong structured signal is present.

**Autocorrelation Reduction Perspective.** Our constraints assign equal magnitude budgets across blocks that are ordered by frequency. Although $\mathcal{F}$ only retains the first half of the DFT coefficients, the Hermitian symmetry (Lemma A.1) ensures that the remaining half is constrained as well. This leads to a spread of DFT magnitudes across frequencies, while still permitting some variance.

From the perspective of circular autocorrelation, defined as:

$$r_{\boldsymbol{x}}[\ell] = \frac{1}{N} \sum_{n=0}^{N-1} x_n \, x_{n-\ell \ (\text{mod } N)}, \qquad \ell = 0, \dots, N-1,$$

the squared magnitudes of the DFT coefficients form a DFT pair with the autocorrelation:

$$r_{\boldsymbol{x}}[\ell] = \frac{1}{N} \sum_{k=0}^{N-1} |\hat{x}_k|^2 \, e^{2\pi i k \ell / N}, \quad \ell = 0, \dots, N-1. \quad (19)$$

This implies that spreading the DFT magnitudes reduces

spatial autocorrelation. A more detailed discussion is provided in Appendix F.

## 5. Latent Optimization with Text-to-Image Generative Models

Building on prior work such as ReNO (Eyring et al., 2024) and MPGR (Hwang et al., 2025), we present experimental results on latent optimization for one-step text-to-image generation. The effect of block size $B$ is discussed in Appendix G, and the analysis on the empirical distributions of spectral magnitudes is in Appendix I.

**Reward Models.** We evaluate using four human-preference-based reward models: *Aesthetic Score* (Schuhmann et al., 2022), a predictor of visual appeal trained on human-rated images; *PickScore* (Kirstain et al., 2023), trained using paired human judgments of text-image outputs; *HPSv2* (Wu et al., 2024), a human preference predictor correlating strongly with subjective evaluations; and *ImageReward* (Xu et al., 2023), trained on expert annotations for text-to-image evaluation. When one of these models is used as the given reward, the others serve as held-out rewards to assess generalization and determine whether the effects transfer across different human-aligned criteria.

**Baselines.** We use prompts from the animal dataset (Black et al., 2024) for Aesthetic Score and T2I-CompBench++ (Huang et al., 2025) for the other reward models following the setup of MPGR (Hwang et al., 2025). We report results using FLUX-schnell (Labs, 2024) as the primary generative model, with additional experiments on SDXL-Turbo, SANA-Sprint, and SD-Turbo in Appendices N and O. We compare our gradient preconditioning method against several regularization-based approaches: KL (Kingma & Welling, 2014), Kurtosis (Shkolnik et al., 2020), ReNO (Eyring et al., 2024), PRNO (Tang et al., 2024), and MPGR (Hwang et al., 2025). Two reference baselines are also included: one without optimization (No Opt.) and one without regularization (No Reg.). In addition, we evaluate two weighting schemes: (i) fixed-weight regularization, where the regularization gradient is scaled only by the coefficient

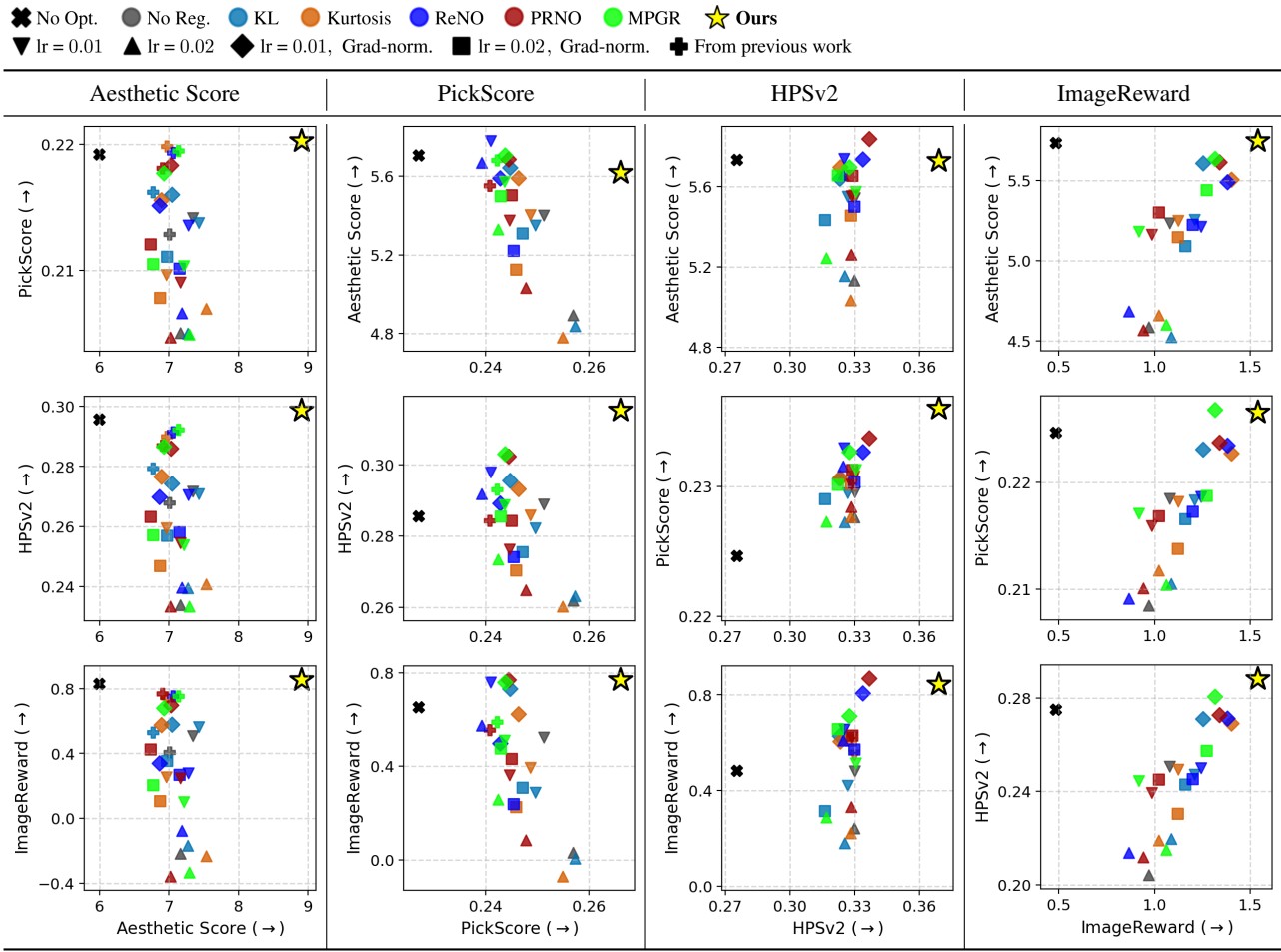

*Figure 2.* **Quantitative results with FLUX model.** Each column corresponds to the same given reward (x-axis), and different held-out rewards (y-axis). Each point denotes the score after 200 iterations, with higher positions and more rightward placement indicating better trade-offs. For baselines, multiple points are plotted across learning rates and regularization schemes. Our method consistently achieves the best trade-off across all reward–held-out reward pairs.

$\lambda$, and (ii) gradient-normalized regularization, where the regularization gradient is rescaled to match the magnitude of the reward gradient to ensure balanced contributions. For Aesthetic Score and PickScore, we also report the results from MPGR (Hwang et al., 2025). Baseline results across various $\lambda$ values are in Appendix J, confirming no value closes the gap.

**Implementation Details.** We initialize each run from white Gaussian noise and optimize for 200 iterations with FLUX and 50 iterations with SDXL-Turbo. The optimization is performed using Adam (Kingma & Ba, 2015) optimizer, with gradient clipping at 0.03 and learning rates of 0.02 (FLUX) and 0.1 (SDXL-Turbo). For all regularization-based methods, we also report results with learning rates 0.01 and 0.02, respectively. We set the regularization coefficient to 2.0 for the baselines. Our method projects both the latent vector and its gradient onto the white Gaussian noise feasible set at every iteration; an ablation isolating each component is provided in Appendix H. All experiments were conducted on a single NVIDIA A6000 GPU, taking approx-

imately 1 minute (FLUX) and 20 seconds (SDXL-Turbo) per 50 iterations, respectively.

**Results.** Quantitative and qualitative results are shown in Figures 2 and 3. According to Figure 2, on *Aesthetic Score*, baselines raise the given reward from 6 to at most 7.5, whereas ours reaches around 9; *PickScore*, *HPSv2*, and *ImageReward* show similar behavior, with our method forming the rightmost frontier. Across all settings, we increase the *given* reward substantially *without* significantly losing *held-out* rewards compared to the unoptimized (No Opt.) reference, indicating improved alignment without degrading realism or image quality.

Looking at the qualitative results in Figure 3, regularization-based methods such as PRNO and MPGR, which penalize spatial correlations, often yield realistic outputs. However, this effect is inconsistent, as seen in the second and third examples in Figure 3. These observations highlight the lack of principled design in soft regularization and reinforce the advantage of our gradient preconditioning. Our method

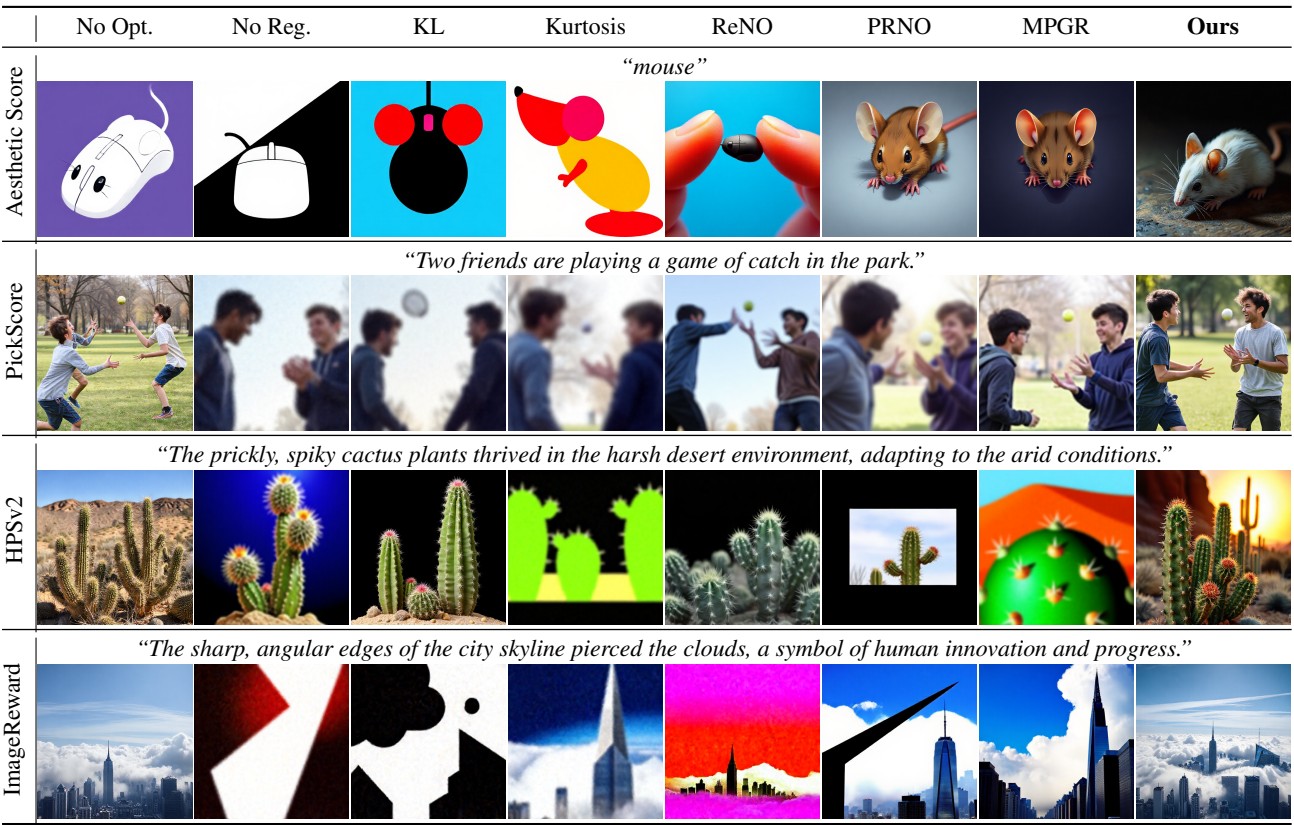

*Figure 3.* **Qualitative results with FLUX model.** Columns denote optimization method; rows correspond to the given reward, with the prompt shown above each row. Our constrained optimization preserves realism and prompt fidelity while attaining higher target scores and strong held-out quality.

*Table 1.* Comparison of FLUX-based reward-guided generation under different iteration counts.

| Method | # Iterations | Aesthetic Score (target) | Pick-Score (held-out) | Wall-clock time (s) |
|---|---|---|---|---|
| No Opt. | 0 | 5.993 | 0.219 | - |
| ReNO | 200 | 7.058 | 0.219 | 232.0 |
| PRNO | 200 | 7.024 | 0.218 | 255.4 |
| MPGR | 200 | 7.133 | 0.220 | 235.5 |
| **Ours (60 iters)** | **60** | 7.120 | 0.220 | **69.7** |
| Ours (200 iters) | 200 | **8.908** | 0.220 | 232.2 |

reliably produces realistic images across all reward models and aligns more faithfully with the text prompt. For example, in the fourth row, it renders the "skyline pierced the clouds", which is not achieved by the unoptimized reference.

**Diversity Analysis.** To verify that our gradient preconditioning does not cause mode collapse, we measure Inception Score (IS) (Salimans et al., 2016) and Vendi Score (Friedman & Dieng, 2023) on 1,125 generated images under Aesthetic Score optimization. Our method (IS: 21.10, Vendi: 6.97) remains within the variance of unoptimized baselines using the same initial latents (IS: 22.33, Vendi: 6.42) and independent latents (IS: 21.57, Vendi: 6.61), confirming no reduction in sample diversity. A detailed diversity analysis including FID is provided in Appendix M.

**Runtime Analysis.** Table 1 reports quantitative results subject to no degradation in the held-out reward (*PickScore*) while optimizing *Aesthetic Score* for each method. Regularization-based baselines reach at most 7.133 after 200 iterations, whereas ours reaches 8.908 under the same iteration budget. This improvement comes with comparable wall-clock time: 200 iterations take about 232–255 seconds for baselines and 232.2 seconds for ours. Moreover, because ours makes faster progress per step, we can also reduce iterations substantially: with only 60 iterations (30% of 200), ours already attains 7.120 in 69.7 seconds, matching the 200-iteration performance of others (7.024–7.133). A best-of-$K$ comparison in Appendix K further demonstrates the advantage of gradient-based optimization.

## 6. Conclusion

We propose gradient preconditioning for efficient and reliable reward-guided generation with one-step generative models. Projecting reward gradients onto a white Gaussian noise feasible set reshapes each update into a noise-aligned direction, driving faster reward ascent while preventing reward hacking without additional hyperparameters. The closed-form $\mathcal{O}(N \log N)$ projection adds negligible overhead, and experiments confirm substantial reward gains without sacrificing sample quality.

## Impact Statement

Our approach can make reward-guided adaptation more practical by avoiding per-reward fine-tuning and lowering computational overhead. However, improved reward optimization may also enable misuse (e.g., optimizing toward harmful objectives) and can amplify biases or blind spots present in the reward model. We therefore recommend that this method should be deployed alongside safety filtering and careful reward-model auditing.

## Acknowledgements

This work was supported by the NRF grant (RS-2026-25486000); IITP grants (RS-2024-00399817, RS-2025-25441313, RS-2025-25443318, RS-2026-25526850) funded by MSIT, Korea; the Industrial Technology Innovation Program grant (RS-2025-02317326) funded by MOTIE, Korea; the InnoCORE program of MSIT (AI Meta-Scientist, N10260110); the National Supercomputing Center with supercomputing resources and technical support (KSC-2025-CRE-0475); the Advanced GPU Utilization Support Program; and the DRB-KAIST SketchTheFuture Research Center.

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

## A. Hermitian Symmetry

**Lemma A.1.** *Let $\boldsymbol{x} \in \mathbb{R}^N$, and let $\hat{\boldsymbol{x}} = \boldsymbol{F}\boldsymbol{x}$ denote its DFT. Then, $\hat{\boldsymbol{x}}$ satisfies the Hermitian symmetry (Proakis & Manolakis, 2007):*

$$\hat{x}_k = \overline{\hat{x}_{N-k}} \quad \text{for } k = 0, \ldots, N-1, \tag{20}$$

*and the coefficients $\hat{x}_0$ and $\hat{x}_{N/2}$ are real-valued.*

*Proof.*

$$\hat{x}_k = \frac{1}{\sqrt{N}} \sum_{j=0}^{N-1} x_j e^{-2\pi \frac{jk}{N} i} = \overline{\frac{1}{\sqrt{N}} \sum_{j=0}^{N-1} x_j e^{2\pi \frac{jk}{N} i}} = \overline{\frac{1}{\sqrt{N}} \sum_{j=0}^{N-1} x_j e^{-2\pi \frac{-jk}{N} i}} = \overline{\hat{x}_{-k \bmod N}} \tag{21}$$

In particular, $\hat{x}_0 = \overline{\hat{x}_0}$ and $\hat{x}_{N/2} = \overline{\hat{x}_{N/2}}$ are real valued. $\qquad\square$

## B. Proof of Proposition 4.1

**Proposition 4.1.** *The mapping $\mathcal{F}$ is a bijection from $\mathbb{R}^N$ to $\mathbb{C}^{N/2}$. Moreover, if $\boldsymbol{z} \sim \mathcal{CN}(\boldsymbol{0}, \boldsymbol{I}_{N/2})$, then $\mathcal{F}^{-1}(\boldsymbol{z}) \sim \mathcal{N}(\boldsymbol{0}, \boldsymbol{I}_N)$.*

*Proof.* Let $N$ be even and let $\boldsymbol{F} \in \mathbb{C}^{N \times N}$ be the unitary DFT. For $\boldsymbol{x} \in \mathbb{R}^N$ write $\hat{\boldsymbol{x}} = \boldsymbol{F}\boldsymbol{x}$ and define $\mathcal{F} : \mathbb{R}^N \to \mathbb{C}^{N/2}$ by

$$y_0 = \frac{\hat{x}_0}{\sqrt{2}} + \frac{\hat{x}_{N/2}}{\sqrt{2}} i, \qquad y_k = \hat{x}_k, \quad k = 1, \ldots, \tfrac{N}{2} - 1. \tag{22}$$

**Bijectivity.** Given $\boldsymbol{y} \in \mathbb{C}^{N/2}$, define $\hat{\boldsymbol{x}} \in \mathbb{C}^N$ by

$$\hat{x}_0 = \sqrt{2}\,\Re(y_0), \quad \hat{x}_{N/2} = \sqrt{2}\,\Im(y_0), \quad \hat{x}_k = y_k, \quad \hat{x}_{N-k} = \overline{y_k} \quad (k = 1, \ldots, \tfrac{N}{2} - 1), \tag{23}$$

and put

$$\boldsymbol{x} = \boldsymbol{F}^\dagger \hat{\boldsymbol{x}} \in \mathbb{R}^N, \tag{24}$$

where $\boldsymbol{F}^\dagger$ denotes the conjugate transpose of $\boldsymbol{F}$. Since $\boldsymbol{F}$ is unitary, we have $\boldsymbol{F}^\dagger = \boldsymbol{F}^{-1}$.

Equation 23 enforces Hermitian symmetry of $\hat{\boldsymbol{x}}$, whence $\boldsymbol{x}$ in equation 24 is real. It is immediate from equation 22–equation 23 that this construction inverts equation 22; hence $\mathcal{F}^{-1}$ exists and is given by equation 23–equation 24. Therefore $\mathcal{F}$ is a bijection.

**Gaussian Preservation.** Let $\boldsymbol{z} \sim \mathcal{CN}(\boldsymbol{0}, \boldsymbol{I}_{N/2})$ with independent elements, construct $\hat{\boldsymbol{x}} \in \mathbb{C}^N$ by

$$\hat{x}_0 = \sqrt{2}\,\Re(z_0), \quad \hat{x}_{N/2} = \sqrt{2}\,\Im(z_0), \quad \hat{x}_k = z_k, \quad \hat{x}_{N-k} = \overline{z_k} \quad (k = 1, \ldots, \tfrac{N}{2} - 1), \tag{25}$$

and set $\boldsymbol{x} = \boldsymbol{F}^\dagger \hat{\boldsymbol{x}} \in \mathbb{R}^N$. Since $\Re(z_0), \Im(z_0) \overset{\text{i.i.d.}}{\sim} \mathcal{N}(0, \tfrac{1}{2})$, $E[z_k] = 0$, $E[z_k \overline{z_\ell}] = \delta_{k\ell}$, and $E[z_k^2] = 0$, $\hat{\boldsymbol{x}}$ is jointly Gaussian and Hermitian-symmetric. We verify the spectral covariance element-wise: variances

$$\mathbb{E}[\hat{x}_0 \overline{\hat{x}_0}] = 1, \qquad \mathbb{E}[\hat{x}_{N/2} \overline{\hat{x}_{N/2}}] = 1, \qquad \mathbb{E}[\hat{x}_k \overline{\hat{x}_k}] = \mathbb{E}[\hat{x}_{N-k} \overline{\hat{x}_{N-k}}] = 1, \tag{26}$$

off-diagonals across different indices vanish by independence, within each conjugate pair $\{k, N-k\}$

$$\mathbb{E}[\hat{x}_k \hat{x}_{N-k}] = \mathbb{E}[z_k^2] = 0, \tag{27}$$

and within the real coefficients is zero by independence,

$$\mathbb{E}[\hat{x}_0 \overline{\hat{x}_{N/2}}] = \mathbb{E}[\hat{x}_{N/2} \overline{\hat{x}_0}] = 0, \tag{28}$$

as are all real-complex cross terms with $k \in \{1, \ldots, \tfrac{N}{2} - 1\}$. Hence

$$\mathbb{E}[\hat{\boldsymbol{x}} \hat{\boldsymbol{x}}^\dagger] = \boldsymbol{I}_N. \tag{29}$$

Unitarity of $\boldsymbol{F}$ gives

$$\text{Cov}(\boldsymbol{x}) = \mathbb{E}\big[\boldsymbol{x}\boldsymbol{x}^\top\big] = \Re\big(\mathbb{E}\big[\boldsymbol{x}\boldsymbol{x}^\dagger\big]\big) = \Re\big(\boldsymbol{F}^\dagger \mathbb{E}\big[\hat{\boldsymbol{x}}\hat{\boldsymbol{x}}^\dagger\big]\boldsymbol{F}\big) = \Re\big(\boldsymbol{F}^\dagger \boldsymbol{I}_N \boldsymbol{F}\big) = \boldsymbol{I}_N. \tag{30}$$

Therefore, $\boldsymbol{x}$ is a real jointly Gaussian vector with diagonal covariance $\boldsymbol{I}_N$. For *real* jointly Gaussian vectors, a diagonal covariance implies independence. Hence, the elements of $\boldsymbol{x}$ are i.i.d. $\mathcal{N}(0,1)$. Equivalently, $\mathcal{F}^{-1}(\boldsymbol{z}) \sim \mathcal{N}(\boldsymbol{0}, \boldsymbol{I}_N)$ with independent elements.

$\square$

## C. Proof of Proposition 4.2

**Proposition 4.2.** *The mapping $\mathcal{F}^{-1}$ is $\mathbb{R}$-linear, and for any $\boldsymbol{z} \in \mathbb{C}^{N/2}$, $\|\mathcal{F}^{-1}(\boldsymbol{z})\|_2^2 = 2\|\boldsymbol{z}\|_2^2$.*

*Proof.* To show the $\mathbb{R}$-linearity of $\mathcal{F}^{-1}$, we first show the $\mathbb{R}$-linearity of $\mathcal{F}$.

**$\mathbb{R}$-linearity of $\mathcal{F}$.** Let $\boldsymbol{x}, \boldsymbol{y} \in \mathbb{R}^N$ and $a, b \in \mathbb{R}$. Write $\hat{\boldsymbol{x}} = \boldsymbol{F}\boldsymbol{x}$ and $\hat{\boldsymbol{y}} = \boldsymbol{F}\boldsymbol{y}$. By linearity of the DFT,

$$\boldsymbol{F}(a\boldsymbol{x} + b\boldsymbol{y}) = a\,\boldsymbol{F}\boldsymbol{x} + b\,\boldsymbol{F}\boldsymbol{y}. \tag{31}$$

Recall the definition of $\mathcal{F}$ (equation 4): for $k = 1, \ldots, \frac{N}{2} - 1$, $(\mathcal{F}(\boldsymbol{x}))_k = \hat{x}_k$, $(\mathcal{F}(\boldsymbol{x}))_0 = \frac{1}{\sqrt{2}}\hat{x}_0 + \frac{1}{\sqrt{2}}\hat{x}_{N/2}\,i$, and analogously for $\boldsymbol{y}$. Each component of $\mathcal{F}$ is a $\mathbb{R}$-linear combination of elements of $\hat{\boldsymbol{x}}$ (or $\hat{\boldsymbol{y}}$). Hence

$$\mathcal{F}(a\boldsymbol{x} + b\boldsymbol{y}) = a\,\mathcal{F}(\boldsymbol{x}) + b\,\mathcal{F}(\boldsymbol{y}). \tag{32}$$

Therefore $\mathcal{F}$ is $\mathbb{R}$-linear.

**$\mathbb{R}$-linearity of $\mathcal{F}^{-1}$.** Since $\mathcal{F}$ is bijective (Proposition 4.1), for arbitrary $\boldsymbol{u}, \boldsymbol{v} \in \mathbb{C}^{N/2}$ there exist unique $\boldsymbol{x}, \boldsymbol{y} \in \mathbb{R}^N$ with $\boldsymbol{u} = \mathcal{F}(\boldsymbol{x})$ and $\boldsymbol{v} = \mathcal{F}(\boldsymbol{y})$. Using the $\mathbb{R}$-linearity of $\mathcal{F}$,

$$\boldsymbol{u} + \boldsymbol{v} = \mathcal{F}(\boldsymbol{x}) + \mathcal{F}(\boldsymbol{y}) = \mathcal{F}(\boldsymbol{x} + \boldsymbol{y}), \qquad a\boldsymbol{u} + b\boldsymbol{v} = \mathcal{F}(a\boldsymbol{x} + b\boldsymbol{y}), \tag{33}$$

for all $a, b \in \mathbb{R}$. Apply $\mathcal{F}^{-1}$ to both equalities to obtain

$$\mathcal{F}^{-1}(a\boldsymbol{u} + b\boldsymbol{v}) = \mathcal{F}^{-1}\big(\mathcal{F}(a\boldsymbol{x} + b\boldsymbol{y})\big) = a\,\boldsymbol{x} + b\,\boldsymbol{y} = a\,\mathcal{F}^{-1}(\boldsymbol{u}) + b\,\mathcal{F}^{-1}(\boldsymbol{v}). \tag{34}$$

Thus $\mathcal{F}^{-1}$ is $\mathbb{R}$-linear.

**Norm Relation.** We first introduce Parseval's identity:

$$\|\boldsymbol{x}\|_2^2 = \|\hat{\boldsymbol{x}}\|_2^2. \tag{35}$$

*Proof of equation 35.* Let $\boldsymbol{F} \in \mathbb{C}^{N \times N}$ be the unitary DFT, so $\boldsymbol{F}^\dagger \boldsymbol{F} = \boldsymbol{I}_N$. Then

$$\|\hat{\boldsymbol{x}}\|_2^2 = \hat{\boldsymbol{x}}^\dagger \hat{\boldsymbol{x}} = (\boldsymbol{F}\boldsymbol{x})^\dagger(\boldsymbol{F}\boldsymbol{x}) = \boldsymbol{x}^\dagger \boldsymbol{F}^\dagger \boldsymbol{F}\boldsymbol{x} = \boldsymbol{x}^\dagger \boldsymbol{x} = \|\boldsymbol{x}\|_2^2. \tag{36}$$

This proves equation 35.

By the definition of $\mathcal{F}$ (cf. equation 4), write $\boldsymbol{y} = \mathcal{F}(\boldsymbol{x})$ from $\hat{\boldsymbol{x}}$ as

$$y_0 = \frac{\hat{x}_0}{\sqrt{2}} + \frac{\hat{x}_{N/2}}{\sqrt{2}}\,i, \qquad y_k = \hat{x}_k \quad (k = 1, \ldots, \tfrac{N}{2} - 1). \tag{37}$$

We now connect $\|\boldsymbol{y}\|_2^2$ and $\|\boldsymbol{x}\|_2^2$ step by step. First, expand $\|\hat{\boldsymbol{x}}\|_2^2$ by pairing the conjugate-symmetric bins:

$$\|\hat{\boldsymbol{x}}\|_2^2 = |\hat{x}_0|^2 + |\hat{x}_{N/2}|^2 + \sum_{k=1}^{N/2-1}\left(|\hat{x}_k|^2 + |\hat{x}_{N-k}|^2\right). \tag{38}$$

Using equation 37,

$$|y_0|^2 = \left| \frac{1}{\sqrt{2}} \hat{x}_0 + \frac{1}{\sqrt{2}} \hat{x}_{N/2} \, i \right|^2 = \frac{1}{2} |\hat{x}_0|^2 + \frac{1}{2} |\hat{x}_{N/2}|^2 \implies |\hat{x}_0|^2 + |\hat{x}_{N/2}|^2 = 2 |y_0|^2. \tag{39}$$

For $k = 1, \dots, \frac{N}{2} - 1$, we have $y_k = \hat{x}_k$, and by Hermitian symmetry $|\hat{x}_{N-k}| = |\hat{x}_k|$, hence

$$|\hat{x}_k|^2 + |\hat{x}_{N-k}|^2 = 2 |y_k|^2 \qquad (k = 1, \dots, \frac{N}{2} - 1). \tag{40}$$

Substituting equation 39 and equation 40 into equation 38 yields

$$\|\hat{\boldsymbol{x}}\|_2^2 = 2 |y_0|^2 + \sum_{k=1}^{N/2-1} 2 |y_k|^2 = 2 \|\boldsymbol{y}\|_2^2. \tag{41}$$

Combining equation 35 and equation 41 gives

$$\|\boldsymbol{x}\|_2^2 = \|\hat{\boldsymbol{x}}\|_2^2 = 2 \|\boldsymbol{y}\|_2^2 \iff \|\boldsymbol{y}\|_2^2 = \frac{1}{2} \|\boldsymbol{x}\|_2^2. \tag{42}$$

**Corollary (Inverse map).** Let $\boldsymbol{z} \in \mathbb{C}^{N/2}$ and set $\boldsymbol{x} = \mathcal{F}^{-1}(\boldsymbol{z})$. Then $\boldsymbol{z} = \mathcal{F}(\boldsymbol{x})$ and equation 42 with $\boldsymbol{y} = \boldsymbol{z}$ gives

$$\|\boldsymbol{x}\|_2^2 = 2 \|\boldsymbol{z}\|_2^2 \qquad \text{i.e.,} \qquad \|\mathcal{F}^{-1}(\boldsymbol{z})\|_2^2 = 2 \|\boldsymbol{z}\|_2^2. \tag{43}$$

$\square$

## D. Projection onto the White Gaussian Noise Feasible Set in the Spectral Domain

We derive an $\mathcal{O}(N \log B)$ algorithm ($N = 2PB$) that computes the closest vector $\dot{\boldsymbol{y}} \in \mathcal{G}_{\mathbb{C}}$ (equation 6) to a given input $\boldsymbol{y} \in \mathbb{C}^{N/2}$ in Euclidean distance.

As edge cases, situations may arise where the projection solution is not uniquely defined. The first occurs when more than 78.5% of the elements in $\boldsymbol{y}^{(p)}$ (i.e., $\frac{\pi}{4} B$ entries) have *exactly equal* magnitudes, and that shared magnitude is the largest within the block. The second occurs when an element has *exactly zero* magnitude. Although both scenarios are exceedingly rare under 64-bit floating-point arithmetic—unless artificially constructed—we address them by perturbing each such $y_j$ with a small complex Gaussian noise $\delta\epsilon$, where $\delta = 10^{-6}$ and $\epsilon \sim \mathcal{CN}(0, 1)$, thereby ensuring that the projection remains well-defined and robust.

Let us now derive the closest vector $\dot{\boldsymbol{y}} \in \mathcal{G}_{\mathbb{C}}$ mathematically.

Given an input $\boldsymbol{y} \in \mathbb{C}^{N/2}$, we aim to solve the following optimization problem:

$$\dot{\boldsymbol{y}} = \operatorname*{argmin}_{\tilde{\boldsymbol{y}} \in \mathbb{C}^{N/2}} \|\tilde{\boldsymbol{y}} - \boldsymbol{y}\|_2^2$$

$$\text{subject to} \quad \|\tilde{\boldsymbol{y}}^{(p)}\|_1 = \frac{\sqrt{\pi}}{2} B, \quad \|\tilde{\boldsymbol{y}}^{(p)}\|_2^2 = B, \quad \text{for all } p = 0, \dots, P-1,$$

where $\dot{\boldsymbol{y}}^{(p)} \in \mathbb{C}^B$ denotes the $p$-th block sub-vector of $\dot{\boldsymbol{y}}$.

To solve this problem, we introduce the following Lagrangian function associated with the constraints:

$$\mathcal{L}(\dot{\boldsymbol{y}}, \lambda_1, \lambda_2) = \frac{1}{2} \|\dot{\boldsymbol{y}} - \boldsymbol{y}\|_2^2 + \sum_{p=0}^{P-1} \lambda_{1,p} \left( \left\| \dot{\boldsymbol{y}}^{(p)} \right\|_1 - \frac{\sqrt{\pi}}{2} B \right) + \sum_{p=0}^{P-1} \lambda_{2,p} \left( \left\| \dot{\boldsymbol{y}}^{(p)} \right\|_2^2 - B \right), \tag{44}$$

To compute the optimality condition, we differentiate the Lagrangian with respect to $\dot{\boldsymbol{y}}$. Let $\boldsymbol{I}_p \in \mathbb{C}^{\frac{N}{2} \times \frac{N}{2}}$ denote a diagonal *block indicator matrix* that selects the $p$-th block of $\dot{\boldsymbol{y}}$, i.e., $[\boldsymbol{I}_p]_{jj} = 1$ if $j \in \{pB, \dots, pB + B - 1\}$ and 0 otherwise. Then, the gradient of the Lagrangian becomes:

$$\nabla_{\dot{\boldsymbol{y}}} \mathcal{L}(\dot{\boldsymbol{y}}, \lambda_1, \lambda_2) = \dot{\boldsymbol{y}} - \boldsymbol{y} + \sum_{p=0}^{P-1} \lambda_{1,p} \boldsymbol{I}_p \nabla_{\dot{\boldsymbol{y}}} \|\dot{\boldsymbol{y}}\|_1 + \sum_{p=0}^{P-1} \lambda_{2,p} \boldsymbol{I}_p \dot{\boldsymbol{y}}. \tag{45}$$

To derive the optimality condition, we set the gradient of the Lagrangian to zero, i.e., $\nabla_{\dot{\boldsymbol{y}}}\mathcal{L}(\dot{\boldsymbol{y}},\lambda_1,\lambda_2)=0$. Let us now focus on the $j$-th coordinate of the $p$-th block, i.e., for indices $j\in\{pB,\ldots,pB+B-1\}$. The first-order condition for each such $j$ becomes:

$$(1+\lambda_{2,p})\,\dot{y}_j+\lambda_{1,p}\nabla_{\dot{y}_j}\,|\dot{y}_j|=y_j. \tag{46}$$

If $\dot{y}_j\neq 0$, the subgradient of the complex $\ell_1$-norm simplifies to:

$$\nabla_{\dot{y}_j}\,|\dot{y}_j|=\frac{\dot{y}_j}{|\dot{y}_j|}, \tag{47}$$

so the optimality condition becomes:

$$(1+\lambda_{2,p})\,\dot{y}_j+\lambda_{1,p}\frac{\dot{y}_j}{|\dot{y}_j|}=y_j. \tag{48}$$

Since the Lagrange multipliers $\lambda_{1,p}$ and $\lambda_{2,p}$ are real-valued, each optimal solution $\dot{y}_j$ can be expressed in polar form as $\dot{y}_j=s_je^{i\theta_j}$, where $s_j\geq 0$ denotes the magnitude and $\theta_j$ the phase. If $y_j\neq 0$, we set $\theta_j=\arg(y_j)$, as the optimality condition equation 48 implies that $\dot{y}_j$ and $y_j$ must share the same phase. If $y_j=0$, then $\theta_j$ can be chosen arbitrarily, since the direction is unconstrained in this case.

To justify that $s_j\geq 0$ holds at the optimum, suppose by contradiction that the optimal solution is $\dot{y}_j=-s_je^{i\theta_j}$ for some $s_j>0$. Then, flipping the sign yields:

$$\left|y_j-(-s_je^{i\theta_j})\right|^2=\left|y_j+s_je^{i\theta_j}\right|^2<\left|y_j-s_je^{i\theta_j}\right|^2, \tag{49}$$

which contradicts the assumption that $\dot{y}_j=-s_je^{i\theta_j}$ is optimal, since both solutions have the same magnitude and thus satisfy the constraints equally well. Therefore, the optimal $s_j$ must be non-negative.

Substituting $\dot{y}_j=s_je^{i\theta_j}$ into equation 48 yields:

$$(1+\lambda_{2,p})\,s_j+\lambda_{1,p}=|y_j|. \tag{50}$$

Otherwise, $\dot{y}_j=0$, we can still express it in polar form as $\dot{y}_j=s_je^{i\theta_j}$ with $s_j=0$ and arbitrary $\theta_j$.

Thus, regardless of whether $\dot{y}_j$ is zero or nonzero, we may uniformly express the solution in polar form as $\dot{y}_j=s_je^{i\theta_j}$ with $s_j\geq 0$. This reformulation allows us to convert the original complex-valued projection problem into a real-valued optimization over the magnitudes $s_j$. For each block indexed by $p$, we solve:

$$\min_{\{s_j\geq 0\}}\sum_{j=pB}^{pB+B-1}(s_j-|y_j|)^2\quad\text{subject to}\quad\sum_{j=pB}^{pB+B-1}s_j=\frac{\sqrt{\pi}}{2}B,\quad\sum_{j=pB}^{pB+B-1}s_j^2=B. \tag{51}$$

To solve this constrained optimization problem, we introduce the Karush–Kuhn–Tucker (KKT) conditions. The corresponding Lagrangian is defined as:

$$\mathcal{L}'(s,\lambda_1,\lambda_2,\boldsymbol{\tau})=\sum_{j=pB}^{pB+B-1}\tfrac{1}{2}(s_j-|y_j|)^2$$
$$+\lambda_1\left(\sum_{j=pB}^{pB+B-1}s_j-\frac{\sqrt{\pi}}{2}B\right)+\lambda_2\left(\sum_{j=pB}^{pB+B-1}s_j^2-B\right)+\sum_{j=pB}^{pB+B-1}\tau_js_j, \tag{52}$$

where $\tau_j\leq 0$ are the KKT multipliers associated with the non-negativity constraints $s_j\geq 0$.

Differentiating with respect to $s_j$ for $j=pB,\ldots,pB+B-1$, and applying the complementary slackness condition yields:

$$s_j-|y_j|+\lambda_1+2\lambda_2s_j+\tau_j=0,\quad\tau_j\leq 0,\quad\tau_js_j=0. \tag{53}$$

First, consider the case $\tau_j<0$. Then, by complementary slackness, it must be that $s_j=0$. Substituting into the stationarity condition gives:

$$\tau_j=|y_j|-\lambda_1<0, \tag{54}$$

which implies $|y_j| < \lambda_1$.

On the other hand, if $\tau_j = 0$, then the stationarity condition simplifies to:

$$(1 + 2\lambda_2)s_j = |y_j| - \lambda_1, \quad s_j \geq 0. \tag{55}$$

Next, we analyze the structure of the optimal solution. Since the constraints enforce both $\sum_j s_j = \frac{\sqrt{\pi}}{2}B$ and $\sum_j s_j^2 = B$, at least two distinct $s_j$'s must be strictly positive.

Also, suppose that there exist two indices $j_1, j_2 \in \{pB, \ldots, pB + B - 1\}$ such that $s_{j_1} > s_{j_2}$ but $|y_{j_1}| < |y_{j_2}|$. Consider swapping their values while preserving all constraints. The change in the objective value would be:

$$\left(s_{j_1} - |y_{j_1}|\right)^2 + \left(s_{j_2} - |y_{j_2}|\right)^2 > \left(s_{j_2} - |y_{j_1}|\right)^2 + \left(s_{j_1} - |y_{j_2}|\right)^2,$$

since the squared error decreases when the larger $s_j$ is matched with the larger $|y_j|$. This contradicts optimality. Hence, the optimal values $s_j$ must preserve the same ordering as the magnitudes $|y_j|$ within each block.

Furthermore, the norm constraints imply that at least two of the $s_j$ values must differ. Let $s_{j_1} > s_{j_2} > 0$, then from the optimality condition:

$$(1 + \lambda_2)(s_{j_1} - s_{j_2}) = (|y_{j_1}| - \lambda_1) - (|y_{j_2}| - \lambda_1) = |y_{j_1}| - |y_{j_2}| \geq 0.$$

This implies $1 + \lambda_2 \geq 0$, and thus

$$(1 + 2\lambda_2)s_j = \max\{|y_j| - \lambda_1, \, 0\} = \mathrm{ReLU}\left(|y_j| - \lambda_1\right), \tag{56}$$

where we used the fact that $s_j = 0$ precisely when $|y_j| < \lambda_1$. This compact expression captures both KKT branches in a unified form.

If $1 + 2\lambda_2 = 0$, then the stationarity condition simplifies to:

$$0 = (1 + 2\lambda_2)s_j = |y_j| - \lambda_1, \tag{57}$$

which implies that all nonzero $s_j$ must satisfy $|y_j| = \lambda_1 = \max_j |y_j|$. Let $\mathcal{I}_{\mathrm{max}}$ be the index set where this holds:

$$\mathcal{I}_{\mathrm{max}} = \left\{j \in \{pB, \ldots, pB + B - 1\} \,\middle|\, |y_j| = \max_{j'} |y_{j'}|\right\}, \quad \text{and let } m = |\mathcal{I}_{\mathrm{max}}|. \tag{58}$$

The two constraints require:

$$\sum_{j \in \mathcal{I}_{\mathrm{max}}} s_j = \frac{\sqrt{\pi}}{2}B, \tag{59}$$

$$\sum_{j \in \mathcal{I}_{\mathrm{max}}} s_j^2 = B. \tag{60}$$

To derive a lower bound on $m$, we apply the Cauchy–Schwarz inequality:

$$\left(\sum_{j \in \mathcal{I}_{\mathrm{max}}} s_j\right)^2 \leq m \cdot \sum_{j \in \mathcal{I}_{\mathrm{max}}} s_j^2. \tag{61}$$

Substituting from equation 59 and equation 60, we obtain:

$$\left(\tfrac{\sqrt{\pi}}{2}B\right)^2 \leq m \cdot B \quad \Rightarrow \quad \tfrac{\pi}{4}B^2 \leq mB \quad \Rightarrow \quad \tfrac{\pi}{4}B \leq m. \tag{62}$$

Thus, in order for the degenerate case $1 + 2\lambda_2 = 0$ to admit a feasible solution, at least $\frac{\pi}{4} \approx 78.5\%$ of the block entries (13 entries when $B = 16$) must have magnitudes exactly equal to $\max_j |y_j|$.

In practice, this is highly unlikely to occur since $y_j$ are continuous values, unless they are artificially set. Nevertheless, to safeguard against this rare but theoretically possible edge case, we introduce a small perturbation to the input:

$$\boldsymbol{y}^{(p)} \leftarrow \boldsymbol{y}^{(p)} + \delta\boldsymbol{\epsilon}, \quad \text{where} \quad \delta = 10^{-6}, \quad \boldsymbol{\epsilon} \sim \mathcal{CN}(\boldsymbol{0}, \boldsymbol{I}), \tag{63}$$

which ensures that ties in magnitude are broken and uniqueness of the solution is preserved under typical 64-bit floating-point arithmetic.

Now, we consider the general case where $1 + 2\lambda_2 \neq 0$. In this case, the optimal magnitudes are given by:

$$s_j = \frac{1}{1 + 2\lambda_2} \cdot \text{ReLU}\left(|y_j| - \lambda_1\right). \tag{64}$$

To enforce the constraints, we define the following two functions:

$$p_1(\lambda_1) := \sum_{j=pB}^{pB+B-1} \text{ReLU}\left(|y_j| - \lambda_1\right), \tag{65}$$

$$p_2(\lambda_1) := \sum_{j=pB}^{pB+B-1} \text{ReLU}\left(|y_j| - \lambda_1\right)^2. \tag{66}$$

Substituting $s_j$ into the constraint equations yields:

$$\sum_j s_j = \frac{1}{1 + 2\lambda_2} \cdot p_1(\lambda_1) = \frac{\sqrt{\pi}}{2} B, \tag{67}$$

$$\sum_j s_j^2 = \frac{1}{(1 + 2\lambda_2)^2} \cdot p_2(\lambda_1) = B. \tag{68}$$

From equation 67 and equation 68, we eliminate $\lambda_2$ to derive the identity:

$$\frac{p_1^2(\lambda_1)}{p_2(\lambda_1)} = \frac{\pi}{4} B. \tag{69}$$

This relation provides an equation of $\lambda_1$. Note that $\frac{p_1^2(\lambda_1)}{p_2(\lambda_1)}$ is a decreasing function with respect to $\lambda_1$ on the range $(-\infty, \max_j |y_j|)$. The value $\frac{p_1^2(\lambda_1)}{p_2(\lambda_1)}$ approaches $B$ as $\lambda_1 \to -\infty$, and approaches $0$ as $\lambda_1 \to \max_j |y_j|$.

For efficient optimal value search, we define $\boldsymbol{w}$ to be the descending-sorted array of magnitudes within the block:

$$\boldsymbol{w} := \text{sort descending}\left(\{\,|y_{pB}|, \ldots, |y_{pB+B-1}|\,\}\right), \quad k = 0, \ldots, B - 1. \tag{70}$$

Define the cumulative sums

$$S_{1,k} := \sum_{l=0}^{k} w_l, \quad S_{2,k} := \sum_{l=0}^{k} w_l^2. \tag{71}$$

Then, for $\lambda^{(k)} \in [w_{k+1}, w_k)$ (with $w_B := -\infty$ for convenience), we compute:

$$p_1\left(\lambda^{(k)}\right) = \sum_{l=0}^{k} \left(w_l - \lambda^{(k)}\right) = S_{1,k} - (k+1)\lambda^{(k)}, \tag{72}$$

$$p_2\left(\lambda^{(k)}\right) = \sum_{l=0}^{k} \left(w_l - \left(\lambda^{(k)}\right)\right)^2 = S_{2,k} - 2\lambda^{(k)} S_{1,k} + (k+1)\left(\lambda^{(k)}\right)^2. \tag{73}$$

Substituting into the constraint $\frac{p_1^2(\lambda^{(k)})}{p_2(\lambda^{(k)})} = \frac{\pi}{4} B$ (letting $\gamma := \frac{\pi}{4}$), we obtain:

$$\frac{p_1^2(\lambda^{(k)})}{p_2(\lambda^{(k)})} = \gamma B, \tag{74}$$

$$p_1^2(\lambda^{(k)}) = \gamma B \cdot p_2(\lambda^{(k)}), \tag{75}$$

$$\left(S_{1,k} - (k+1)\lambda^{(k)}\right)^2 = \gamma B \cdot \left(S_{2,k} - 2\lambda^{(k)}S_{1,k} + (k+1)\left(\lambda^{(k)}\right)^2\right). \tag{76}$$

Rearranging yields the quadratic equation:

$$(k+1)(\gamma B - k - 1)\left(\lambda^{(k)}\right)^2 - 2S_{1,k}(\gamma B - k - 1)\lambda^{(k)} + \gamma B S_{2,k} - S_{1,k}^2 = 0. \tag{77}$$

To ensure real roots, we evaluate the discriminant:

$$\Delta = S_{1,k}^2(\gamma B - k - 1)^2 - (k+1)(\gamma B - k - 1)\left(\gamma B S_{2,k} - S_{1,k}^2\right) \tag{78}$$

$$= (\gamma B - k - 1)\left[S_{1,k}^2(\gamma B - k - 1) - (k+1)(\gamma B S_{2,k} - S_{1,k}^2)\right] \tag{79}$$

$$= (\gamma B - k - 1)\left[\gamma B S_{1,k}^2 - (k+1)\gamma B S_{2,k}\right] \tag{80}$$

$$= \gamma B(\gamma B - k - 1)\left(S_{1,k}^2 - (k+1)S_{2,k}\right) \tag{81}$$

$$= \gamma B(k + 1 - \gamma B)\left((k+1)S_{2,k} - S_{1,k}^2\right). \tag{82}$$

We clarify that the term $(k+1)S_{2,k} - S_{1,k}^2$ is nonnegative by the Cauchy–Schwarz inequality:

$$(S_{1,k})^2 = \left(\sum_{\ell=0}^{k} w_\ell\right)^2 \leq (k+1)\sum_{\ell=0}^{k} w_\ell^2 = (k+1)S_{2,k}. \tag{83}$$

Therefore, real-valued solutions exist only when $k + 1 \geq \gamma B$. Otherwise, the solution falls under the degenerate case previously handled in equation 62.

Thus, the solution for $\lambda^{(k)}$ is given by:

$$\lambda^{(k)} = \frac{S_{1,k} \pm \frac{\sqrt{\Delta}}{k+1-\gamma B}}{k+1}, \tag{84}$$

$$= \frac{S_{1,k} - \frac{\sqrt{\Delta}}{k+1-\gamma B}}{k+1} \quad \left(\text{since } \frac{S_{1,k}}{k+1} \geq w_k\right), \tag{85}$$

$$= \frac{S_{1,k}}{k+1} - \frac{\sqrt{\gamma B}}{k+1}\sqrt{\frac{(k+1)S_{2,k} - S_{1,k}^2}{k+1-\gamma B}}. \tag{86}$$

If $w_k > \lambda^{(k)} \geq w_{k+1}$, then this $\lambda^{(k)}$ is the solution $\lambda_1$ for equation 69. Once $\lambda_1$ is determined, the corresponding $\lambda_2$ is recovered from equation 67.

The final solution is given by:

$$s_j = \frac{1}{1 + 2\lambda_2}\text{ReLU}\left(|y_j| - \lambda_1\right) \tag{87}$$

$$= \frac{\sqrt{\pi}B}{2p_1(\lambda_1)}\text{ReLU}\left(|y_j| - \lambda_1\right) \quad \text{(by equation 67)}, \tag{88}$$

and

$$\dot{y}_j = \frac{\sqrt{\pi}B}{2p_1(\lambda_1)}\text{ReLU}\left(|y_j| - \lambda_1\right)e^{i\theta_j} \tag{89}$$

where $\theta_j$ is $\arg(y_j)$ if $y_j \neq 0$, and a random angle otherwise.

In practice, exact zero $y_j$ in 64-bit floating-point representations are extremely rare unless artificially introduced. Nonetheless, when this occurs, we perturb $y_j$ by adding a small white Gaussian noise $\delta\epsilon$, where $\delta = 10^{-6}$ and $\epsilon \sim \mathcal{CN}(0, 1)$.

## D.1. Computational Perspective

From a computational standpoint, the dominant cost arises from sorting the magnitudes $|y_j|$ within each block, which takes $\mathcal{O}(B \log B)$ time. Since there are $P$ blocks in total, the overall sorting cost is $\mathcal{O}(N \log B)$. All subsequent operations—such as computing prefix sums, evaluating the discriminant, and updating the filtered vector—can be performed in linear time per block. Therefore, the total runtime of the algorithm is $\mathcal{O}(N \log B)$.

Moreover, because each block is processed independently, the entire procedure is naturally parallelizable and well-suited for GPU acceleration.

## E. Range of the Magnitude Spectrum

In this subsection, we analyze the range of $|y_j|$. Since the constraints are permutation-invariant, we consider $|y_0|$ without loss of generality. The extreme cases occur when $|y_0|$ is the largest among the $B$ elements in the same block, and the remaining $(B-1)$ elements have equal magnitude. Let $a := |y_0|$ and $b := |y_1| = |y_2| = \cdots = |y_{B-1}|$. Under these assumptions, the following constraints hold (with $\gamma := \frac{\pi}{4}$):

$$a + (B-1)b = \sqrt{\gamma}B, \qquad a^2 + (B-1)b^2 = B. \tag{90}$$

Solving for $b$ in terms of $a$, we have:

$$b = \frac{\sqrt{\gamma}B - a}{B - 1}. \tag{91}$$

Substituting this into the second equation yields:

$$a^2 + (B-1)\left(\frac{\sqrt{\gamma}B - a}{B - 1}\right)^2 = B, \tag{92}$$

which simplifies to:

$$a^2 - 2\sqrt{\gamma}a + \gamma B - B + 1 = 0. \tag{93}$$

Solving this quadratic in $a$, we obtain:

$$a = \sqrt{\gamma} + \sqrt{(1 - \gamma)(B - 1)}, \quad b = \sqrt{\gamma} - \sqrt{\frac{1 - \gamma}{B - 1}}. \tag{94}$$

Thus, the extreme values that $|y_0|$ can attain under these constraints are:

$$|y_0|_{\min} = \sqrt{\gamma} - \sqrt{(1 - \gamma)(B - 1)}, \tag{95}$$

$$|y_0|_{\max} = \sqrt{\gamma} + \sqrt{(1 - \gamma)(B - 1)}. \tag{96}$$

For block size $B = 16$, we find $|y_0|_{\min} < 0$ and $|y_0|_{\max} \approx 2.68$. Therefore, $|y_j|$ lies in the range $[0.0, 2.68]$, which corresponds to approximately 99.92% coverage under the $\chi_2/\sqrt{2}$ distribution.

## F. Relationship between Flat Magnitude Spectrum and Zero Autocorrelation

Let $x \in \mathbb{R}^N$ and let $\hat{x} = Fx$ be the *unitary* DFT. Define the *circular* (period-$N$) autocorrelation

$$r_x[\ell] = \frac{1}{N} \sum_{n=0}^{N-1} x_n \, x_{n-\ell \,(\text{mod } N)}, \qquad \ell = 0, \ldots, N-1. \tag{97}$$

According to discrete Wiener–Khinchin relations (Khintchine, 1934), the periodogram $|\hat{x}_k|^2$ and the circular autocorrelation form an exact DFT pair:

$$|\hat{x}_k|^2 = \sum_{\ell=0}^{N-1} r_x[\ell] \, e^{-2\pi i k\ell/N}, \quad k = 0, \ldots, N-1, \tag{98}$$

$$r_x[\ell] = \frac{1}{N} \sum_{k=0}^{N-1} |\hat{x}_k|^2 \, e^{2\pi i k\ell/N}, \quad \ell = 0, \ldots, N-1. \tag{99}$$

Taking expectations of both sides yields the Wiener–Khinchin relations in expectation:

$$\mathbb{E}\big[|\hat{x}_k|^2\big] = \sum_{\ell=0}^{N-1} \mathbb{E}[r_{\boldsymbol{x}}[\ell]] \; e^{-2\pi i k \ell/N}, \quad k = 0, \ldots, N-1, \tag{100}$$

$$\mathbb{E}[r_{\boldsymbol{x}}[\ell]] = \frac{1}{N} \sum_{k=0}^{N-1} \mathbb{E}\big[|\hat{x}_k|^2\big] \; e^{2\pi i k \ell/N}, \quad \ell = 0, \ldots, N-1. \tag{101}$$

Let $c := \mathbb{E}[\, r_{\boldsymbol{x}}[0] \,] = \frac{1}{N} \mathbb{E}\big[\|\boldsymbol{x}\|_2^2\big]$. Then by equation 101,

$$\mathbb{E}[r_{\boldsymbol{x}}[\ell]] \;=\; c\, \delta_N[\ell] \quad \Longleftrightarrow \quad \mathbb{E}\big[|\hat{x}_k|^2\big] \equiv c, \tag{102}$$

where $\delta_N[\ell] = 1$ if $\ell \equiv 0 \pmod{N}$ and $0$ otherwise. Hence a flat expected magnitude spectrum is *identical* to zero expected autocorrelation at all nonzero circular lags.

For a single finite realization, $|\hat{x}_k|^2$ fluctuates around its expectation, so perfect flatness is not observed. Nevertheless, promoting a well-spread (near-flat) magnitude distribution across frequencies reduces off-origin correlations via equation 99.

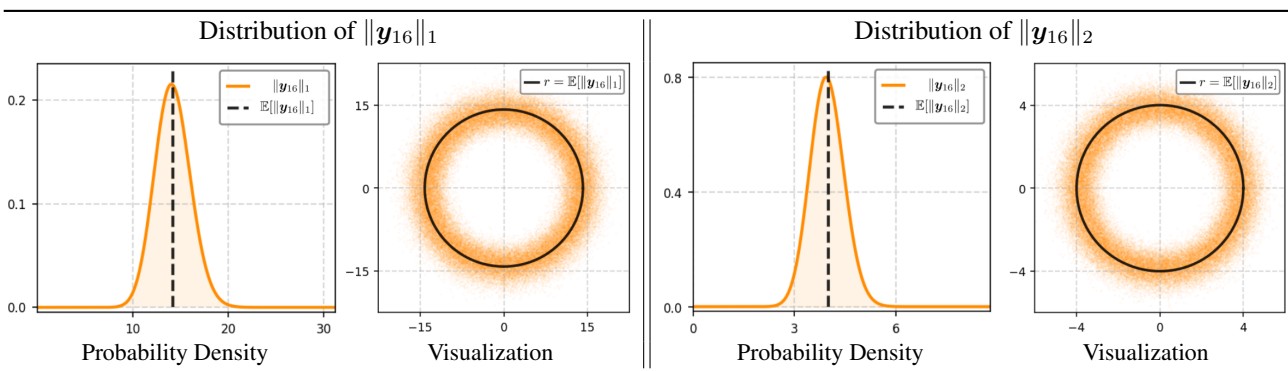

*Figure 4.* **Distributions of $\ell_1$ and $\ell_2$ norms of $\boldsymbol{y}_{16} \sim \mathcal{CN}(\boldsymbol{0}, \boldsymbol{I}_{16})$.** The radial plots visualize 100K samples, where radius indicates the norm and angle indicates $\arg(\boldsymbol{1}^\top \boldsymbol{y}_{16})$. In both cases, the norms are concentrated around their expected values.

## G. Effect of Block Size

In this section, we discuss the effect of the block size $B$ and present related experimental results.

Our constraints match the blockwise $\ell_1$ and $\ell_2$ norms to their theoretical expectations under the Gaussian distribution. As the block size $B$ decreases, the variance of these blockwise norms increases, and the feasible set moves farther away from the set of white Gaussian noise. Conversely, as $B$ increases, the blockwise norms of white Gaussian noise concentrate around their expectations, and the feasible set becomes closer to white Gaussian noise. At the same time, however, the feasible set becomes *less tight* for larger $B$, since the number of constraints decreases with the number of blocks.

This trade-off can also be seen from the allowable magnitude range derived in Section E. For example, the maximum allowed value of $|y_j|$ is 2.11 when $B = 8$, but increases to 84.74 when $B = 32,768$. This illustrates how the block size controls the tightness of the constraints around the notion of white noise: smaller blocks enforce tighter local control, while larger blocks permit much larger deviations in individual coefficients.

Based on this analysis, we set $B = 16$, which yields a feasible set that remains close to white Gaussian noise (the minimum cosine similarity with 1M samples from $\mathcal{N}(\boldsymbol{0}, \boldsymbol{I}_N)$ with $N = 65,536$ is 0.988), while still imposing sufficiently tight constraints on white Gaussian noise. The theoretical distributions of the blockwise $\ell_1$ and $\ell_2$ norms are shown in Figure 4.

*Table 2.* Effect of block size $B$ on FLUX-based reward-guided generation.

| Block size $B$ | MPGR (baseline) | 8 | **16** | 32 | 64 | 32,768 |
|---|---|---|---|---|---|---|
| Aesthetic Score (target) | 7.1329 | 8.5883 | 8.9078 | **9.0833** | 8.1558 | 7.8834 |
| PickScore (held-out) | 0.2195 | 0.2177 | 0.2203 | 0.2193 | 0.2161 | 0.2110 |
| HPSv2 (held-out) | 0.2922 | 0.2940 | 0.2986 | 0.3111 | 0.2828 | 0.2623 |

We further evaluate the effect of $B$ on FLUX (Labs, 2024) when optimizing the Aesthetic Score (Schuhmann et al., 2022). Table 2 reports the target reward (Aesthetic Score) and held-out rewards (PickScore and HPSv2) for different block sizes. For all $B$, our method improves the Aesthetic Score over MPGR (Hwang et al., 2025). However, the held-out rewards degrade as $B$ becomes very large, reflecting the looser nature of the constraints. In contrast, the performance for $B = 8$, 16, and 32 is comparable in both target and held-out metrics, indicating that the method is not overly sensitive around our chosen value $B = 16$.

## H. Ablation Study: Gradient and Latent Projection

Our method applies two projections at each iteration: the reward gradient is projected onto the white Gaussian noise feasible set before the optimizer step (gradient projection), and the latent vector is projected after each update (latent projection). To isolate the contribution of each component, we ablate by using gradient projection alone (without latent projection) and compare it against the full method. Table 3 reports results under Aesthetic Score optimization on FLUX.

Gradient projection alone already achieves an Aesthetic Score of $8.6469$, far exceeding regularization-based baselines ($7.1329$ for MPGR) and confirming that gradient preconditioning is the primary driver of reward improvement. Adding latent projection further increases the target score to $8.9078$ and substantially improves held-out rewards across all metrics, most notably ImageReward ($0.7678 \rightarrow 0.8548$). This indicates that latent projection stabilizes the optimization trajectory, keeping the latent vector noise-like throughout optimization and thereby better preserving image quality.

*Table 3.* Ablation of gradient and latent projection under Aesthetic Score optimization on FLUX.

| Method | Aesthetic Score ($\uparrow$) | PickScore (held-out, $\uparrow$) | HPSv2 (held-out, $\uparrow$) | ImageReward (held-out, $\uparrow$) |
|---|---|---|---|---|
| No Opt. | 5.9932 | 0.2192 | 0.2958 | 0.8299 |
| No Reg. | 7.0072 | 0.2129 | 0.2679 | 0.4076 |
| PRNO | 7.0244 | 0.2183 | 0.2861 | 0.6997 |
| MPGR | 7.1329 | 0.2195 | 0.2922 | 0.7538 |
| Ours (grad. proj. only) | 8.6469 | 0.2176 | 0.2894 | 0.7678 |
| **Ours** (grad. + latent proj.) | **8.9078** | 0.2203 | 0.2986 | 0.8548 |

## I. Magnitude Distributions Across Optimization Methods

To complement the quantitative statistics, we further examine how different latent optimization methods affect the empirical distributions of spectral magnitudes. For a fixed latent sample, we compute the elementwise magnitudes $|y_i|$ as well as the blockwise norms $\|\boldsymbol{y}^{(p)}\|_1$ and $\|\boldsymbol{y}^{(p)}\|_2$ of the complex-valued spectrum. For each method, we plot these quantities for both a white Gaussian noise and the optimized latent vector. For our method, since the blockwise norms are fixed, we instead show the empirical probability density function of $|y_i|$.

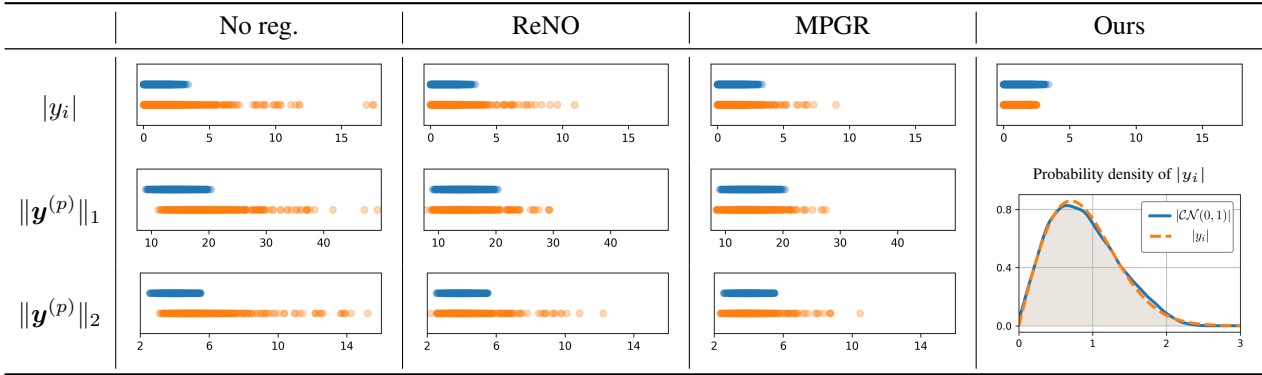

*Figure 5.* Empirical distributions of spectral magnitudes for each optimization method. ■ indicates values obtained from a white Gaussian noise, and ■ indicates values from the optimized latent vector. For each frequency or block $p$, the corresponding magnitude or norm is plotted at its location on the horizontal axis. The rightmost panel shows the empirical and theoretical probability density of $|y_i|$.

Compared to the unregularized optimization, ReNO and MPGR reduce the deviation of the empirical magnitudes from those of the white Gaussian noise. However, the $\ell_1$ distributions of MPGR still exhibit noticeable deviations, indicating that soft penalties on $\|\boldsymbol{y}^{(p)}\|_1$ do not fully prevent drift in blockwise magnitudes. Moreover, the regularization-based methods produce individual frequencies whose magnitudes are substantially larger than expected under the white Gaussian noise. In contrast, our gradient preconditioning keeps elementwise magnitudes around the Gaussian distribution, and the upper bound is set (Appendix E). This behavior is consistent with the notion of white noise discussed in Section 4.5, where no single frequency component is disproportionately strong.

*Table 4.* Effect of regularization coefficient $\lambda$ on FLUX under Aesthetic Score optimization. Our method requires no $\lambda$.

| Method | $\lambda$ | Aesthetic Score ($\uparrow$) | PickScore (held-out, $\uparrow$) | HPSv2 (held-out, $\uparrow$) |
|---|---|---|---|---|
| ReNO | 0.5 | 7.1902 | 0.2142 | 0.2714 |
| ReNO | 1.0 | 7.1931 | 0.2149 | 0.2765 |
| ReNO | 2.0 | 6.8614 | 0.2152 | 0.2698 |
| ReNO | 4.0 | 7.2284 | 0.2191 | 0.2887 |
| PRNO | 0.5 | 6.9051 | 0.2109 | 0.2606 |
| PRNO | 1.0 | 6.9507 | 0.2148 | 0.2762 |
| PRNO | 2.0 | 7.0244 | 0.2183 | 0.2861 |
| PRNO | 4.0 | 6.9942 | 0.2203 | 0.2979 |
| MPGR | 0.5 | 7.1044 | 0.2146 | 0.2769 |
| MPGR | 1.0 | 6.9868 | 0.2144 | 0.2759 |
| MPGR | 2.0 | 6.9257 | 0.2177 | 0.2866 |
| MPGR | 4.0 | 7.0487 | 0.2210 | 0.2934 |
| **Ours** | **N/A** | **8.9078** | 0.2203 | 0.2986 |

*Table 5.* Comparison of gradient-based optimization and best-of-$K$ sampling on FLUX under Aesthetic Score optimization.

| Method | Aesthetic Score ($\uparrow$) |
|---|---|
| No Opt. | 5.9932 |
| max@60 | 6.5268 |
| max@200 | 6.6513 |
| MPGR (200 iters) | 7.1329 |
| **Ours (60 iters)** | 7.1200 |
| **Ours (200 iters)** | **8.9078** |

*Table 6.* Results at 1,000-prompt scale on T2I-CompBench++ with PickScore as target reward on FLUX.

| Method | PickScore (target, $\uparrow$) | Aesthetic Score (held-out, $\uparrow$) | HPSv2 (held-out, $\uparrow$) |
|---|---|---|---|
| No Opt. | 0.2270 | 5.7060 | 0.2856 |
| PRNO | 0.2444 | 5.6856 | 0.3023 |
| MPGR | 0.2438 | 5.7056 | 0.3031 |
| **Ours** | **0.2660** | 5.6161 | **0.3152** |

## J. Sensitivity to Regularization Coefficient

All regularization-based methods require a coefficient $\lambda$ that balances reward and regularization gradients. To verify that the performance gap is not an artifact of the specific value $\lambda = 2.0$ used in the main experiments, we sweep $\lambda \in \{0.5, 1.0, 2.0, 4.0\}$ for ReNO, PRNO, and MPGR on FLUX under Aesthetic Score optimization. Our method requires no $\lambda$, as gradient preconditioning replaces soft regularization entirely. Table 4 reports results for all settings.

Baselines vary across $\lambda$ values (ReNO: 6.86–7.23, PRNO: 6.91–7.02, MPGR: 6.93–7.10), but even the best $\lambda$ per method (ReNO: 7.23, PRNO: 7.02, MPGR: 7.10) does not approach our result of 8.91. This confirms that the performance gap is not due to a suboptimal choice of $\lambda$ for the baselines.

## K. Comparison to Best-of-$K$ Sampling

Best-of-$K$ (max@$K$) sampling selects the highest-reward image from $K$ independent random samples without gradient computation. Table 5 compares max@$K$ for $K \in \{60, 200\}$ against gradient-based methods on FLUX under Aesthetic Score optimization.

Max@$K$ shows limited scalability: max@200 gains only 0.12 over max@60 (6.65 vs. 6.53), and both remain well below any gradient-based method. Even MPGR at 200 iterations (7.13) substantially outperforms max@200 (6.65), confirming that gradient information provides a qualitatively different level of optimization. Among gradient-based methods, our approach achieves the strongest results: with only 60 iterations we already match MPGR at 200 iterations (7.12 vs. 7.13), and with 200 iterations we reach 8.91, outperforming all baselines by a large margin.

## L. Large-Scale Experiment

To assess whether results hold at a larger prompt scale, we run an additional experiment using 1,000 prompts from T2I-CompBench++ (Huang et al., 2025) with PickScore as the target reward on FLUX. We compare against the two strongest regularization-based baselines. Table 6 reports the target PickScore and two held-out rewards.

Our method achieves the highest target PickScore (0.2660 vs. 0.2444) and the highest held-out HPSv2 (0.3152 vs. 0.3031), while the held-out Aesthetic Score remains close to baselines (5.62 vs. 5.69–5.71). The same tendencies observed in the main paper hold at this larger scale.

## M. Diversity Analysis

We provide a more detailed analysis of sample diversity under Aesthetic Score optimization with FLUX, complementing the IS and Vendi Score results in the main paper. Tables 7 and 8 report IS (Salimans et al., 2016), Vendi Score (Friedman & Dieng, 2023), and FID on 1,125 generated images. Two unoptimized baselines are included: one sharing the same initial latents as our method (same init.) and one drawn from entirely independent latents (new init.). Our method remains within the variance of both baselines on IS, and its Vendi Score slightly increases, confirming no diversity loss.

To further characterize the distribution shift, we compute FID between each pair of settings. The intra-baseline FID (14.49) reflects natural variation between two independent sets of unoptimized images. The FID between our optimized outputs and the same-init. baseline (26.80) is consistent with the independent-baseline FID (26.35), indicating that the distribution shift is systematic rather than a result of overfitting to specific initial latents. Combined with IS and Vendi Score, this confirms that our method does not suffer from mode collapse and its distribution shift reflects a genuine reward-guided improvement.

*Table 7.* IS and Vendi Score on 1,125 images.

| Setting | IS ($\uparrow$) | Vendi ($\uparrow$) |
|---|---|---|
| No Opt. (same init.) | $22.3259 \pm 1.4144$ | 6.4248 |
| No Opt. (new init.) | $21.5689 \pm 1.2859$ | 6.6103 |
| **Ours** | $21.1043 \pm 1.7549$ | 6.9682 |

*Table 8.* FID between pairs of settings.

| Comparison | FID ($\downarrow$) |
|---|---|
| Same init. vs. new init. | 14.4947 |
| Same init. vs. Ours | 26.7967 |
| New init. vs. Ours | 26.3463 |

## N. Experimental Results with SANA-Sprint and SD-Turbo

We evaluate our method on two additional one-step generative models to confirm generalization beyond FLUX-schnell and SDXL-Turbo. Tables 9 and 10 report results under Aesthetic Score optimization on SANA-Sprint (Chen et al., 2025) and SD-Turbo (Sauer et al., 2024), respectively. On SANA-Sprint, our method achieves an Aesthetic Score of 8.16, outperforming the best baseline ReNO (7.61) by 0.55 points while held-out rewards remain comparable. On SD-Turbo, our method reaches 8.57, surpassing baselines (7.52–7.58) by approximately 1.0 point with held-out PickScore and HPSv2 on par. Combined with FLUX-schnell and SDXL-Turbo in the main paper, gradient preconditioning generalizes consistently across four distinct one-step architectures.

*Table 9.* Results on SANA-Sprint under Aesthetic Score optimization.

| Method | Aesthetic Score ($\uparrow$) | Pick-Score ($\uparrow$) | HPSv2 ($\uparrow$) | Image-Reward ($\uparrow$) |
|---|---|---|---|---|
| No Opt. | 6.7899 | 0.2245 | 0.3094 | 1.0558 |
| ReNO | 7.6137 | 0.2249 | 0.3093 | 1.0331 |
| PRNO | 7.5928 | 0.2243 | 0.3080 | 1.0716 |
| MPGR | 7.5687 | 0.2248 | 0.3099 | 1.0371 |
| **Ours** | **8.1635** | 0.2248 | 0.3084 | 1.0397 |

*Table 10.* Results on SD-Turbo under Aesthetic Score optimization.

| Method | Aesthetic Score ($\uparrow$) | Pick-Score ($\uparrow$) | HPSv2 ($\uparrow$) | Image-Reward ($\uparrow$) |
|---|---|---|---|---|
| No Opt. | 4.1926 | 0.1832 | 0.1633 | $-2.2794$ |
| ReNO | 7.5566 | 0.2194 | 0.2798 | 0.1743 |
| PRNO | 7.5841 | 0.2189 | 0.2763 | 0.2056 |
| MPGR | 7.5189 | 0.2193 | 0.2773 | 0.2384 |
| **Ours** | **8.5735** | 0.2193 | 0.2768 | 0.1950 |

# O. Experimental Results with SDXL-Turbo

We report quantitative and qualitative results with SDXL-Turbo in Figures 6 and 7, respectively. Across all reward models, our method consistently achieves better trade-offs than the baselines while not significantly losing held-out rewards compared to the unoptimized outputs (No Opt.), thereby improving reward alignment without compromising realism or image quality. By contrast, baselines across different learning rates and regularization schemes fail to achieve our trade-offs. We reason this limitation arises from soft regularization, which provides no guarantee of white Gaussian noise properties even with sophisticated loss formulations.

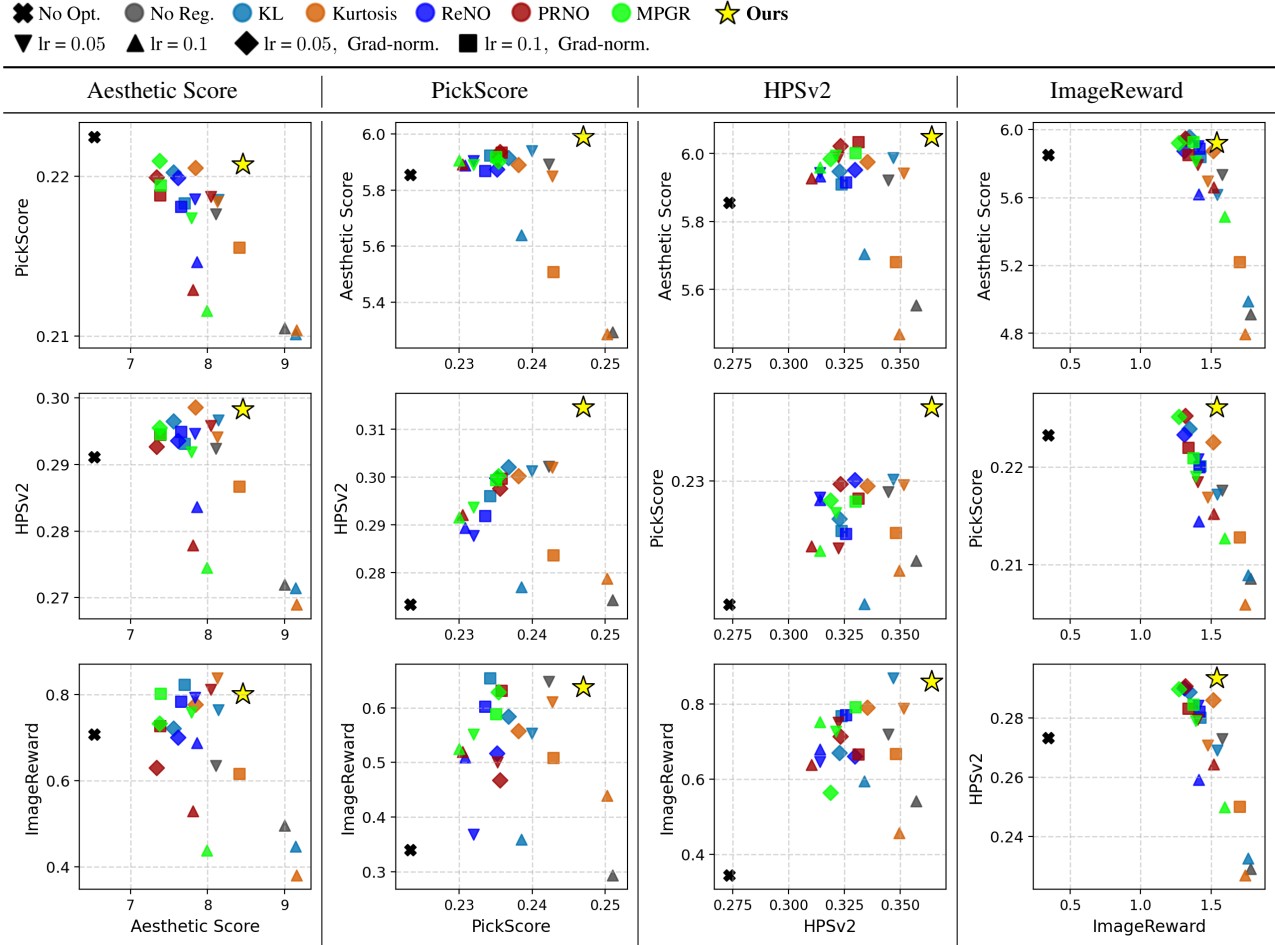

*Figure 6.* **Quantitative Results with SDXL-Turbo.** Each column corresponds to the same given reward (x-axis), and different held-out rewards (y-axis). Each point denotes the score after 50 iterations, with higher positions and more rightward placement indicating better trade-offs. For baselines, multiple points are plotted across learning rates and regularization schemes.

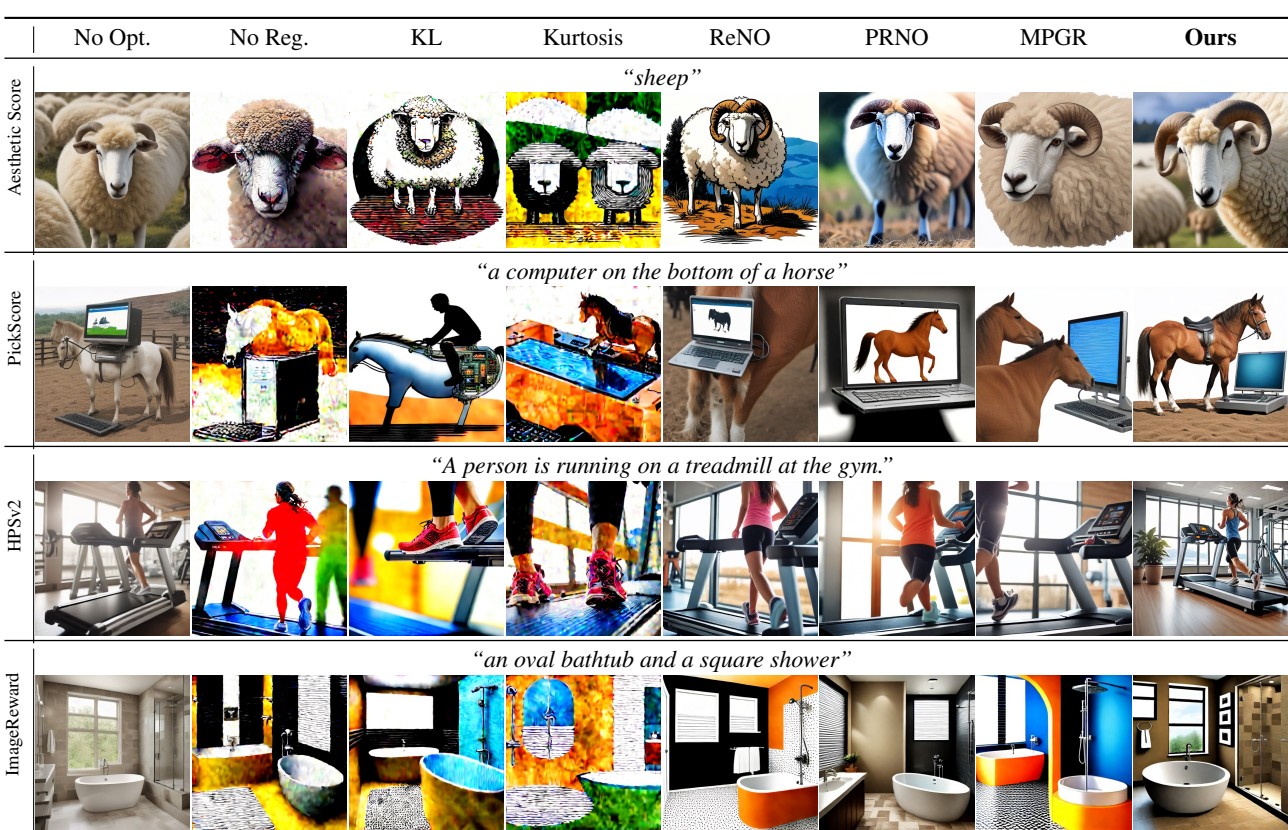

*Figure 7.* **Qualitative results with SDXL-Turbo.** Columns denote optimization method; rows correspond to the given reward, with the prompt shown above each row. Our constrained optimization preserves realism and prompt fidelity while attaining higher target scores and strong held-out quality.

