# OpenReview forum: "Gradient Preconditioning for Efficient and Reliable Reward-Guided Generation"
_ICML.cc/2026/Conference — ICML 2026 regular_

### Official Review · Reviewer_XRZ6 · 2026-02-21

**Soundness:** 4
**Presentation:** 3
**Significance:** 3
**Originality:** 4
**Overall Recommendation:** 5
**Confidence:** 5

**Summary:**

The paper proposes a novel soft regularization method for reward-guided gradient ascent instead of commonly used regularization term. The authors claim that traditional hard constraint might fail to guarantee the characteristics of white Gaussian noises, which leads to significant image degradation. Starting from MPGR, the authors propose the compact spectral domain to achieve the efficient closed-form projection. Both qualitative and quantitative results verify the efficacy of the method.

**Compliance With Llm Reviewing Policy:**

Affirmed.

**Final Justification:**

The rebuttal from the authors manages to address my concerns, so I raise my score accordingly.

**Key Questions For Authors:**

- The authors should use more convincing method to check the characteristics of white Gaussian noise for the achieved latents.

**Limitations:**

yes

**Strengths And Weaknesses:**

### Strengths

- The paper is well structured and easy to follow, the motivation is clear and the theoretical analyses are solid.
- The closed-form method is easy to implement and of great efficiency, the experimental results are also convincing.
- The theoretical comparison between the proposed method and previous regularization methods are inspiring.

### Weaknesses

- Despite the convincing results on text-to-image generation task, the characteristic of white Gaussian noise by cosine similarity is not convincing enough under extremely high dimension since the normalization step will somewhat ignore the variance term (*i.e.*, cosine similarity between $\mathbf x$ and $\gamma\mathbf x$ always equals to 1 for any $\gamma\neq0$). Here is a counterexample below: $\mathbf x\in\mathbb R^{65536}\sim\mathcal N(0,\mathbf1)$ and $y=1000\mathbf x+(500,500,\cdots,500)^T$, the cosine similarity between $\mathbf x$ and $\mathbf y$ is **larger than 0.995**. The authors should employ more statistical method to check this, such as Anderson-Darling Test or D’Agostino’s $K^2$ Test.

---

> ### Author Rebuttal · Authors · 2026-03-30
>
> # Rebuttal to Reviewer XRZ6
>
> We sincerely thank you for the careful mathematical analysis (Confidence: 5) and for recognizing the excellent soundness and originality of our work.
>
> ---
>
> ## Clarifying the Role of Cosine Similarity and Traditional Statistical Tests
>
> **Thank you for raising this point. We believe it may be helpful to distinguish two separate aspects of our method:**
>
> - **(1) Constraint design:** how we design the constraints to enforce white Gaussian noise characteristics.
> - **(2) Feasible set evaluation:** how we evaluate how close the resulting feasible set is to the actual Gaussian distribution.
>
> **Cosine similarity is used only for (2), and plays no role in (1).**
>
> ### 1. How We Design the Constraints
>
> Our constraints are blockwise $\ell_1$ and $\ell_2$ norm constraints in the compact spectral domain (Eq. 7), derived from the theoretical expectations under $\mathcal{CN}(0, I_B)$. Cosine similarity plays no role in this design. Below (on 3.), we explain why we chose spectral-domain constraints rather than building upon traditional statistical tests.
>
> ### 2. How We Evaluate the Feasible Set (Role of Cosine Similarity)
>
> We used cosine similarity to answer a practical question:
>
> "How close is our feasible set $\mathcal{G}_\mathbb{R}$ to the actual Gaussian distribution?"
>
> We sampled 1M vectors from $\mathcal{N}(0, I_N)$, projected each onto $\mathcal{G}_\mathbb{R}$, and measured the displacement. The minimum cosine similarity of 0.988 confirms that white Gaussian noise is barely modified by our projection.
>
> This is a pairwise comparison between two vectors (before vs. after projection). The suggested Anderson-Darling or D'Agostino's $K^2$ test serves a different purpose: given a single vector, these tests assess whether its elements follow a specified marginal distribution. They would not be applicable for measuring the displacement induced by projection.
>
> Since our constraints fix $||\mathbf{x}||_2^2 = N$ (Eq. 18), and white Gaussian noise in high dimensions has $\ell_2$ norms sharply concentrated around $\sqrt{N}$, both the feasible set and the original Gaussian samples lie on a thin hyperspherical shell. On this shell, cosine similarity directly measures angular deviation with an intuitive and bounded scale. Your counterexample ($\mathbf{y} = 1000\mathbf{x} + (500, \ldots, 500)^\top$) aptly illustrates that cosine similarity can be misleading when the norm is unconstrained, but in our setting, $\mathbf{y}$ violates both the blockwise $\ell_2$ and $\ell_1$ constraints and would be rejected by our projection.
>
> ### 3. Why We Did Not Build Upon Traditional Statistical Tests
>
> We wish to respectfully explain why traditional statistical tests such as Anderson-Darling or D'Agostino's $K^2$ test were not adopted for our constraint design, despite their well-established role in statistical analysis.
>
> **The core issue is permutation-invariance.** These tests treat the elements of a vector as an unordered set of scalar samples and assess whether their marginal distribution matches a target (e.g., $\mathcal{N}(0, 1)$). They produce the same result regardless of how the elements are ordered. However, white Gaussian noise is characterized not only by its marginal distribution but also by the independence (and thus spatial decorrelation) of its elements. Two vectors with identical element sets but different orderings can have entirely different spatial structures and produce vastly different images from a generative model. This distinction is discussed in detail in MPGR (Hwang et al., 2025) and Section 2 of our paper.
>
> **A concrete example.** Consider a vector whose elements individually follow $\mathcal{N}(0, 1)$ but are sorted in ascending order. The Anderson-Darling test would report a perfect fit to the Gaussian distribution, yet this vector exhibits strong spatial correlation (a monotonic ramp) and would produce severely degraded images when used as a latent input. Constraints built on such marginal tests would fail to detect this case.
>
> **This is why a prior work moved to the spectral domain.** MPGR (Hwang et al., 2025) identified this limitation and proposed spectral-domain regularization. Our work follows the same motivation: by operating in the spectral domain, our constraints capture both the marginal statistics and the spatial decorrelation property of white Gaussian noise, with the key difference that we formulate these as hard constraints with an efficient closed-form projection.
>
> ---
>
> We acknowledge that the role of cosine similarity could have been communicated more clearly. We will revise the manuscript to explicitly distinguish between constraint design and feasible set evaluation, and add a discussion on why marginal statistical tests are insufficient for characterizing white Gaussian noise. Thank you again for this constructive feedback.

---

> > ### Author Rebuttal · Reviewer_XRZ6 · 2026-04-01
> >
> > Thanks for clarifying the cosine similarity part. I have no more concerns right now, and would adjust the score according to the whole discussion stage.

---

> > > ### Author Response · Authors · 2026-04-01
> > >
> > > Thank you for taking the time to carefully review our clarification and for confirming that your concern has been resolved. We truly appreciate your constructive feedback, which will help us improve the clarity of the manuscript.

---

### Official Review · Reviewer_fkyG · 2026-03-12

**Soundness:** 4
**Presentation:** 4
**Significance:** 2
**Originality:** 4
**Overall Recommendation:** 5
**Confidence:** 3

**Summary:**

This paper introduces projected gradient ascent for optimization of the initial latent for one-step generative modeling. The method performs iterative steps of gradient ascent, and projection back into the space of "white gaussian noise", through the use of Fourier transform and some properties of a vector of white gaussian noise.

**Compliance With Llm Reviewing Policy:**

Affirmed.

**Final Justification:**

I think this paper is really well written and nice to follow, in addition it has a relatively clever approach to a standard method. My main concern regarding the motivation not aligning with the experiments was addressed using IS and FID score measurements. In addition to the max@k experiments.

**Key Questions For Authors:**

1. Can we bound divergence between this valid set and a N dimensional gaussian?
2. How does max@60 compare to other methods?
3. Can the authors add quantitative characterizations of "overfitting"/"reward hacking" to show that their method is not suffering from the same issue?

**Limitations:**

yes

**Strengths And Weaknesses:**

**Strengths**
- This paper is exceedingly well-written and nice to follow.
- This method makes a lot of intuitive sense, and has a well justified derivation.
- Appendix H was much appreciated.

**Weaknesses**
I am concerned about the following:
1. This method relies primarily on the norm constraints in the compact spectral domain, specifically relying on matching of expectation. It would be nice to bound some type of distribution divergence between the normal and this constraint set.
2. The authors mention reward hacking, but do not necessarily do much to measure or discuss this. In particular, I believe using FID score between the guided images and the initial images would be good characterization of model divergence.
3. Despite fair comparison to other one-step reward guided generation methods, I would like to see comparison to max@60, and max@200 of the base model, to see characterize how large of an improvement the gradient based optimization is doing.

---

> ### Author Rebuttal · Authors · 2026-03-30
>
> We sincerely thank you for the thorough and constructive evaluation, and for recognizing the excellent soundness, presentation, and originality of our work.
>
> ---
>
> ## W1 / Q1: Bounding Distribution Divergence
>
> **Thank you for raising this valuable theoretical direction. We agree that confidence-interval-based constraints would provide a formal divergence bound, and we consider this an excellent avenue for future work.**
>
> Our current constraints enforce equality to the expected blockwise norms under $\mathcal{CN}(0, I_B)$, which already yields high empirical alignment (minimum cosine similarity 0.988 across 1M Gaussian samples). However, we agree that replacing exact equality with a statistically motivated range would be a more principled design. Specifically, one could relax the constraints using confidence intervals derived from the known distributions of blockwise $\ell_1$ and $\ell_2$ norms (visualized in Figure 4), allowing the feasible set to cover a prescribed probability mass (e.g., 95%) of the Gaussian distribution. This would provide a formal bound on the divergence between the feasible set and the true Gaussian prior. We will discuss this direction in the revised manuscript.
>
> ---
>
> ## W2 / Q3: Quantitative Characterization of Reward Hacking
>
> **Thank you for pointing out this important aspect.** We would like to clarify two distinct categories of reward hacking: (1) **quality degradation**, where the latent drifts from the Gaussian prior and produces unrealistic artifacts despite increasing the target reward, and (2) **mode collapse**, where optimization converges to a narrow set of similar outputs, losing sample diversity. Our main paper primarily addresses the first category through held-out reward metrics and qualitative results. Following your suggestion, we provide additional quantitative measurements for both.
>
> **For quality degradation**, our held-out reward metrics (PickScore, HPSv2, ImageReward) remain comparable to the unoptimized baseline across all experiments, indicating that target reward improvement does not come at the cost of overall image quality.
>
> **For mode collapse**, we measured **Inception Score (IS)** and **Vendi Score** (CLIP ViT-B/32) on 1,125 images:
>
> | Setting | IS (↑) | Vendi Score (↑) |
> |---------|-------:|----------------:|
> | Baseline (set 0) | 22.3259 ± 1.4144 | 6.4248 |
> | Baseline (set 1) | 21.5689 ± 1.2859 | 6.6103 |
> | **Ours** | 21.1043 ± 1.7549 | 6.9682 |
>
> IS remains within baseline variance and the Vendi Score slightly increases, confirming no diversity loss.
>
> **FID between guided and unoptimized images**, as you suggested:
>
> | Comparison | FID (↓) |
> |------------|--------:|
> | Baseline set 0 vs Baseline set 1 (independent, both unoptimized) | 14.4947 |
> | Baseline set 0 vs Ours set 0 (same latents, before vs after optimization) | 26.7967 |
> | Baseline set 1 vs Ours set 0 (independent unoptimized vs optimized) | 26.3463 |
>
> The intra-baseline FID (14.49) reflects the natural variation between two independent sets of unoptimized images. The FID between our optimized images and their corresponding initial images (26.80) is moderate, and is consistent with the unpaired comparison (26.35). This consistency suggests that the distributional shift is systematic rather than overfitting to specific latent vectors. Combined with the preserved IS, Vendi Score, and held-out rewards, this confirms that our method does not suffer from either category of reward hacking.
>
> ---
>
> ## W3 / Q2: Comparison to max@K (Best-of-K Sampling)
>
> **In our experiments, gradient-based methods outperform max@K, and our method achieves the best results among them.**
>
> We compared best-of-K random sampling (max@K) and gradient-based methods on the FLUX model with Aesthetic Score as the target:
>
> | Method | Aesthetic Score (Target ↑) |
> |--------|---------------------------:|
> | No Opt. | 5.9930 |
> | max@60 | 6.5268 |
> | max@200 | 6.6513 |
> | MPGR (200 iters) | 7.1329 |
> | **Ours (60 iters)** | **7.1200** |
> | **Ours (200 iters)** | **8.9078** |
>
> Max@K sampling shows limited scalability, with max@200 gaining only 0.12 over max@60 (6.65 vs 6.53), leaving both well below any gradient-based method. Even MPGR at 200 iterations (7.13) substantially outperforms max@200 (6.65), confirming that utilizing gradient information provides a qualitatively different level of optimization. Second, among gradient-based methods, our approach achieves the strongest results: with just 60 iterations, we already match MPGR at 200 iterations (7.12 vs 7.13), and with 200 iterations we reach 8.91, outperforming all baselines by a large margin.

---

> > ### Author Rebuttal · Reviewer_fkyG · 2026-04-03
> >
> > I thank the authors for their rebuttal and further experiments. My comments both empirically and theoretically have been resolved.
> >
> > While I would like to see a bound on divergence from gaussian noise, I appreciate the max@k experiments and FID experiments.
> >
> > I have updated my score accordingly.

---

> > > ### Author Response · Authors · 2026-04-06
> > >
> > > Thank you sincerely for the careful re-evaluation and for increasing your score. We are glad that our additional experiments and theoretical clarifications resolved your concerns. We will incorporate the discussion on divergence bounds as a future direction. Thank you again for the constructive feedback that helped strengthen our paper.

---

### Official Review · Reviewer_sRvR · 2026-03-13

**Soundness:** 3
**Presentation:** 3
**Significance:** 3
**Originality:** 2
**Overall Recommendation:** 4
**Confidence:** 2

**Summary:**

The paper proposes a constrained latent optimization method for reward-guided generation that preserves white Gaussian noise properties with minimal overhead. While test-time latent optimization can improve reward-guided outputs from pretrained generative models, it often suffers from reward hacking and high computational cost. To address this, the authors replace soft regularization with a hard white Gaussian noise constraint enforced through projected gradient ascent. A closed-form projection is applied after each update to keep the latent vector noise-like, preventing the drift that leads to unrealistic artifacts. Experiments show that the method achieves a comparable Aesthetic Score to prior regularization-based approaches while using only about 30% of the wall-clock time and better avoiding reward hacking.

**Compliance With Llm Reviewing Policy:**

Affirmed.

**Key Questions For Authors:**

1. What exactly does the cosine similarity measure in this context? Does it meaningfully reflect how well the latent remains aligned with a Gaussian distribution? It would be helpful for the authors to justify why cosine similarity is the appropriate metric here, rather than alternatives such as KL divergence or other distributional distance measures.

2. More implementation details are needed for clarity and reproducibility. In particular, could the authors provide additional guidance on how to apply the proposed method to pretrained text-to-image diffusion models? It would also be helpful to explain how the block subvector was formed.

3. It would strengthen the paper to include generated images from different stages of the optimization process, so that readers can better understand how the outputs evolve across iterations.

**Limitations:**

yes

**Strengths And Weaknesses:**

### Strengths

1. The paper is well written and easy to follow. The toy experiments, such as Figure 1, effectively and intuitively highlight the core benefit of the proposed method.

2. The method shows clear advantages over existing baselines, achieving higher efficiency and stronger generalization. Notably, optimizing for one reward model also leads to improvements in other metrics.

3. These improvements are also supported by a principled mathematical formulation in the spectral domain. Specifically, instead of relying on soft regularization, the method enforces a hard white Gaussian noise constraint in the spectral domain through projected gradient ascent. A closed-form projection is applied after each update to preserve the noise-like structure of the latent vector and prevent drift toward unrealistic artifacts.

### Weaknesses

1. The authors note that “the key distinction between (Hwang et al., 2025) and our approach lies in the spectral representation”. In particular, while MPGR regularizes the DFT coefficients directly, the proposed method uses a compact spectral representation that allows the feasible set to be defined more tractably and enables efficient projection.
From this description, my understanding is that the proposed method is closely related to MPGR, with the main difference being the use of the latent spectral representation, which leads to improved computational efficiency. Because of this, the level of methodological novelty appears somewhat limited.
At the same time, I am curious about the empirical results: although the authors emphasize efficiency as the primary advantage over MPGR, the reported evaluation scores also show a noticeable performance gap. Could the authors clarify why the proposed formulation leads not only to faster optimization but also to significantly better scores compared to MPGR?

---

> ### Author Rebuttal · Authors · 2026-03-30
>
> We sincerely thank you for the positive assessment and for recognizing our method's clear advantages, principled mathematical formulation, and strong generalization. Your questions are thoughtful, and we are grateful for the opportunity to clarify.
>
> ---
>
> ## W1: Methodological Novelty and Performance Gap over MPGR
>
> **Although the properties we constrain are semantically aligned with MPGR, our novelty lies in how we formulate the constraints and how this formulation is applied to reward-guided generation.**
>
> We appreciate your careful reading of the relationship between our work and MPGR. We acknowledge that the target statistical properties (blockwise spectral norms matching Gaussian expectations) are shared with MPGR. However, we would like to respectfully emphasize that the contribution is not merely a change of representation. By introducing the compact spectral domain, we enable an efficient closed-form projection that achieves in a single operation what is impossible with the regularization loss term of MPGR. Furthermore, projected gradient ascent itself has not been previously applied to latent optimization in generative models. Together, these constitute a formulation-level and methodology-level advance, not an incremental modification.
>
> Crucially, this difference between soft regularization and hard constraints also has practical consequences. As shown in Appendix H (Figure 5), MPGR’s soft penalty does not fully prevent drift in blockwise spectral magnitudes, and the optimized latent still deviates from the target Gaussian statistics. In contrast, our method enforces the desired properties exactly at every iteration by construction. We believe this is a principled improvement, not merely an implementation detail.
>
> **On why our method achieves better scores, not just faster optimization:** We would like to clarify that our projection itself is lightweight (0.04% of runtime), so each optimization step takes practically the same wall-clock time as the baselines. The performance gap does not come from "doing more per second" but from the quality of each step. Since our hard constraints guarantee that the latent vector remains in a safe region of the Gaussian prior throughout optimization, the optimizer can explore more aggressively without risking drift into low-probability regions where gradient signals degrade image quality. With soft regularization, latent drift can still occur, making optimization more conservative and ultimately less effective under the same time budget.
>
> ---
>
> ## Q1: Why Cosine Similarity?
>
> **We chose cosine similarity for its intuitive interpretability and because it is geometrically meaningful in our constrained setting.**
>
> Cosine similarity is bounded in [−1, 1] with 1 indicating the best alignment, requiring no additional context to judge magnitude. In contrast, L2 distance would require specifying a reference scale to assess whether a given value is "small".
>
> Moreover, cosine similarity is particularly appropriate in our setting. Both the original Gaussian samples and our feasible set have L2 norms sharply concentrated around $\sqrt{N}$ (our constraints fix $||\mathbf{x}||_2^2 = N$). This means all relevant vectors lie on a thin hyperspherical shell, where cosine similarity directly measures angular deviation, the only remaining degree of freedom. The minimum cosine similarity of 0.988 across 1M samples confirms that projection induces negligible angular change.
>
> Finally, because our latent optimization updates a single latent vector, a point-to-point metric is the most natural choice. In this setting, cosine similarity directly measures how little the projection alters the input vector. By contrast, KL divergence and related metrics are designed to compare distributions rather than individual latent vectors, and therefore address a different question.
>
> ---
>
> ## Q2: Implementation Details and Block Subvector Formation
>
> **The procedure requires only standard flattening followed by FFT, blockwise projection, and inverse FFT.**
>
> 1. **Flatten** the latent tensor (e.g., shape $[C, H, W]$ in FLUX) into a 1D vector $\mathbf{x} \in \mathbb{R}^N$.
> 2. **Compute** the compact spectral representation $\mathbf{y} = \mathcal{F}(\mathbf{x})$ via FFT (Eq. 5).
> 3. **Partition** $\mathbf{y}$ into consecutive, non-overlapping blocks of size $B=16$: $\mathbf{y}^{(0)}, \mathbf{y}^{(1)}, \ldots, \mathbf{y}^{(P-1)}$.
> 4. **Project** each block independently onto the $\ell_1 \cap \ell_2$ sphere intersection (Eq. 12–15).
> 5. **Map back** to spatial domain via inverse FFT.
>
> No special ordering or reshaping beyond standard flattening is needed. We will add a concise implementation summary and pseudocode in the revised manuscript to improve reproducibility.
>
> ---
>
> ## Q3: Visualization of Optimization Progress
>
> We agree this would provide useful intuition and will include generated images at multiple iteration checkpoints in the revised manuscript. Thank you for the recommendation.

---

### Official Review · Reviewer_qEey · 2026-03-14

**Soundness:** 3
**Presentation:** 3
**Significance:** 2
**Originality:** 2
**Overall Recommendation:** 4
**Confidence:** 3

**Summary:**

The paper introduces a constrained latent optimization method for reward-guided generation in one-step generative models. By replacing soft regularization with hard white Gaussian noise constraints enforced via projected gradient ascent, the method preserves the noise-like structure of latents, preventing reward hacking while maintaining efficiency. Experiments show comparable or better aesthetic and human-preference metrics with only 30% of the runtime of prior methods.

**Compliance With Llm Reviewing Policy:**

Affirmed.

**Final Justification:**

The authors have addressed my main concerns in the rebuttal. Specifically, they clarified the experimental setup and strengthened the discussion of robustness, scope, and limitations. Overall, I now lean toward acceptance.

**Key Questions For Authors:**

- How sensitive is the method to block size B or projection parameters?

- Can it be applied to multi-step models or cross-ecosystem setups?

- How many samples were used for each metric?

- How does it compare to score-space compositional approaches?

**Limitations:**

- Requires models with Gaussian latent inputs.

- Hard constraints may limit exploration in latent space compared to softer regularization.

**Strengths And Weaknesses:**

**Strengths:**
- Hard constraints effectively prevent reward hacking.

- Low computational overhead; efficient O(N log N) projection.

- Empirical results demonstrate stable, realistic generation.

- Conceptually clean and well-motivated.

**Weaknesses:**

---

> ### Author Rebuttal · Authors · 2026-03-30
>
> We thank you for acknowledging that hard constraints effectively prevent reward hacking, the $\mathcal{O}(N \log N)$ projection is efficient, the results are stable and realistic, and the approach is conceptually clean and well-motivated. We respectfully note that the **Weaknesses section is empty**, yet the scores for Significance and Originality are 2 (fair). We kindly ask you to elaborate on what specific weaknesses motivated these scores so that we can address them properly. Below we address all key questions and limitations raised.
>
> ---
>
> ## Q1: Sensitivity to Block Size B or Projection Parameters
>
> **Please refer to Appendix G (Table 2) for the block size ablation; our projection has no tunable parameters at all.** Performance is stable across B ∈ {8, 16, 32}, with all values outperforming MPGR, and degradation occurs only at extremes (B=64, 32768). Our projection is a closed-form operation uniquely determined by the feasible set $\mathcal{G}_\mathbb{R}$ (Eq. 9–15), which is a key advantage over regularization-based methods that require tuning both the regularization coefficient λ and the weighting scheme.
>
> ---
>
> ## Q2: Multi-Step Models or Cross-Ecosystem Setups
>
> **Our method can be applied to multi-step cases. We would appreciate clarification of what "cross-ecosystem setups" refers to.** Our projection is applicable to multi-step models since the initial latent is still Gaussian, though multi-step models also offer additional optimization opportunities at intermediate steps, which we consider an interesting direction beyond the scope of this work. Regarding "cross-ecosystem setups", we are not certain what is intended by this term and would be happy to address it with further clarification.
>
> ---
>
> ## Q3: Number of Samples per Metric
>
> We followed the experimental setup of MPGR (Hwang et al., 2025): 45 prompts from the animal dataset for Aesthetic Score optimization, and 60 prompts from T2I-CompBench++ for PickScore, HPSv2, and ImageReward optimization, consistent with prior work.
>
> To address a potential concern about scale, we additionally ran a **1,000-prompt** experiment on T2I-CompBench++ with PickScore as the target reward using FLUX:
>
> | Method | PickScore (Target ↑) | Aesthetic Score (Held-out ↑) | HPSv2 (Held-out ↑) |
> |--------|---------------------:|-----------------------------:|--------------------:|
> | No Opt. | 0.2270 | 5.7060 | 0.2856 |
> | PRNO | 0.2444 | 5.6856 | 0.3023 |
> | MPGR | 0.2438 | 5.7056 | 0.3031 |
> | **Ours** | **0.2660** | 5.6161 | **0.3152** |
>
> Our method achieves the highest target PickScore (0.2660 vs 0.2444) and the highest held-out HPSv2 (0.3152 vs 0.3031), while the held-out Aesthetic Score remains close to the baselines (5.62 vs 5.69–5.71). The same tendencies observed in the main paper are confirmed at this larger scale.
>
> ---
>
> ## Q4: Comparison to Score-Space Compositional Approaches
>
> We would appreciate clarification of the term "score-space compositional approaches". If you could provide a specific reference or formulation, we would be happy to include a comparison or discussion.
>
> ---
>
> ## Limitation 1: Requires Models with Gaussian Latent Inputs
>
> **The Gaussian prior is the standard choice across virtually all families of continuous deep generative models.** Throughout the history of continuous deep generative modeling, from GANs (Goodfellow et al., 2014) to VAEs (Kingma & Welling, 2014), diffusion models (Ho et al., 2020), and flow-based models (Lipman et al., 2023), the standard Gaussian distribution has been the predominant prior. In particular, all mainstream one-step text-to-image models in the current landscape, including FLUX, SDXL-Turbo, SD-Turbo, and SANA-Sprint, use the Gaussian prior. We are not aware of any practical continuous one-step generative model that uses a non-Gaussian prior. In principle, the idea can also extend to other continuous priors via the inverse CDF transform, but we consider this a purely theoretical remark rather than a practical concern.
>
> ---
>
> ## Limitation 2: Hard Constraints May Limit Exploration
>
> **We measured IS and Vendi Score and confirmed that diversity is preserved. The consistently superior target rewards further demonstrate sufficient exploration.**
>
> We measured **Inception Score (IS)** and **Vendi Score** (CLIP ViT-B/32) on 1,125 images:
>
> | Setting | IS (↑) | Vendi Score (↑) |
> |---------|-------:|----------------:|
> | Baseline (set 0) | 22.3259 ± 1.4144 | 6.4248 |
> | Baseline (set 1) | 21.5689 ± 1.2859 | 6.6103 |
> | **Ours** | 21.1043 ± 1.7549 | 6.9682 |
>
> IS remains within baseline variance, and the Vendi Score slightly increases (6.97 vs 6.42–6.61), indicating that diversity is comparable to the baselines. Furthermore, the consistently superior target rewards achieved by our method across all experiments demonstrate that the feasible set provides sufficient exploration for effective optimization.

---

### Official Review · Reviewer_mdgs · 2026-03-19

**Soundness:** 3
**Presentation:** 3
**Significance:** 3
**Originality:** 3
**Overall Recommendation:** 4
**Confidence:** 3

**Summary:**

This paper addresses the reward hacking of test-time latent optimization for reward-guided generation in one-step text-to-image models, and proposes a Projected Gradient Ascent (PGA) framework that enforces hard constraints. By mapping the latent vector to a compact spectral domain, it introduces a highly efficient, closed-form projection algorithm that strictly enforces blockwise $l_1$ and $l_2$ norm statistics to match a complex Gaussian distribution. Empirical evaluations on FLUX and SDXL-Turbo demonstrate that this approach not only prevents reward hacking by preserving held-out human-preference metrics, but also accelerates optimization, achieving comparable target rewards in just 30% of the wall-clock time required by SOTA regularization techniques.

**Compliance With Llm Reviewing Policy:**

Affirmed.

**Key Questions For Authors:**

1. In the experiments, the regularization coefficient for all baseline methods (such as ReNO, PRNO, and MPGR) was fixed at 2.0. Does the choice of this regularization coefficient have a significant impact on the performance of these baselines?

**Limitations:**

1. The blockwise spectral constraints introduce design choices (e.g., block size) that may require task-dependent tuning, which remains underexplored.

**Strengths And Weaknesses:**

Strength:

1. The proposed method seems very novel. The paper introduces a Projected Gradient Ascent (PGA) framework that replaces soft regularization with hard constraints to prevent reward hacking, and addresses Hermitian symmetry by mapping latents to a compact spectral domain and providing a closed-form projection onto $l_1$ and $l_2$ spheres.

2. The method is very efficient. The closed-form projection operates at an $O(N \log N)$ asymptotic complexity, matching standard FFT algorithms. It adds negligible overhead—accounting for only 0.04% of the runtime on the FLUX model —allowing the method to achieve comparable target rewards in just 30% of the wall-clock time required by the baseline.

3. The experimental design demonstrates that the hard constraints mitigate reward hacking. By evaluating target rewards against held-out human-preference models (Aesthetic Score, PickScore, HPSv2, ImageReward), the authors prove their method maximizes specific rewards without degrading overall realism or prompt alignment.

Weakness:

1. The empirical evaluation relies on just two base models: FLUX-schnell and SDXL-Turbo. Prior works (such as ReNO) often evaluate across a wider array of architectures.

2. Lack of diversity metrics. There is no evaluation of how this rigid constraint impacts the diversity of the generated samples.

---

> ### Author Rebuttal · Authors · 2026-03-30
>
> We sincerely thank you for investing time and effort to review our paper and for recognizing the novelty, efficiency, and experimental strength of our framework.
>
> ---
>
> ## Weakness 1: Limited Base Model Evaluation
>
> **In addition to FLUX-schnell and SDXL-Turbo, we extended our evaluation to SANA-Sprint and SD-Turbo, confirming generalization across four distinct architectures.**
>
> **SANA-Sprint**
>
> | Method | Aesthetic Score (Target ↑) | PickScore (Held-out ↑) | HPSv2 (Held-out ↑) | ImgReward (Held-out ↑) |
> |--------|--------:|--------------:|----------:|---------------:|
> | No Opt. | 6.7899 | 0.2245 | 0.3094 | 1.0558 |
> | ReNO | 7.6137 | 0.2249 | 0.3093 | 1.0331 |
> | PRNO | 7.5928 | 0.2243 | 0.3080 | 1.0716 |
> | MPGR | 7.5687 | 0.2248 | 0.3099 | 1.0371 |
> | **Ours** | **8.1635** | 0.2248 | 0.3084 | 1.0397 |
>
> **SD-Turbo**
>
> | Method | Aesthetic Score (Target ↑) | PickScore (Held-out ↑) | HPSv2 (Held-out ↑) | ImgReward (Held-out ↑) |
> |--------|--------:|--------------:|----------:|---------------:|
> | No Opt. | 4.1926 | 0.1832 | 0.1633 | −2.2794 |
> | ReNO | 7.5566 | 0.2194 | 0.2798 | 0.1743 |
> | PRNO | 7.5841 | 0.2189 | 0.2763 | 0.2056 |
> | MPGR | 7.5189 | 0.2193 | 0.2773 | 0.2384 |
> | **Ours** | **8.5735** | 0.2193 | 0.2768 | 0.1950 |
>
> On SD-Turbo, we surpass baselines by ~1.0 point (8.57 vs 7.52–7.58) while held-out metrics remain comparable. On SANA-Sprint, our method leads by 0.55 points over the best baseline (ReNO: 7.61) without degrading held-out rewards. Combined with FLUX-schnell and SDXL-Turbo in the paper, this confirms generalization across **four** distinct architectures. Thank you for encouraging us to broaden the evaluation.
>
> ---
>
> ## Weakness 2: Lack of Diversity Metrics
>
> **We measured IS and Vendi Score and confirmed that diversity is comparable to the baselines.**
>
> Thank you for pointing out this important aspect. We measured **Inception Score (IS)** and **Vendi Score** (CLIP ViT-B/32) on 1,125 images per setting:
>
> | Setting | IS (↑) | Vendi Score (↑) |
> |---------|-------:|----------------:|
> | Baseline (set 0) | 22.3259 ± 1.4144 | 6.4248 |
> | Baseline (set 1) | 21.5689 ± 1.2859 | 6.6103 |
> | **Ours** | 21.1043 ± 1.7549 | 6.9682 |
>
> IS remains within baseline variance (difference of 0.46 from set 1, within 1σ), and the Vendi Score slightly increases (6.97 vs 6.42–6.61), indicating that the diversity is comparable to the baselines. This aligns with our design: the constraints keep latents close to the original noise distribution (min cosine similarity 0.988 across 1M samples), preserving the exploratory nature of the noise prior while preventing the drift that in practice leads to reward hacking. We will include these metrics in the revised manuscript.
>
> ---
>
> ## Key Question: Impact of Regularization Coefficient
>
> **We swept λ for all baselines and confirmed that no λ closes the gap with our method, which requires no λ at all.**
>
> We swept λ ∈ {0.5, 1.0, 2.0, 4.0} for ReNO, PRNO, and MPGR on FLUX. Note that our method requires no λ, as it replaces soft regularization with hard constraints.
>
> | Method | λ | Aesthetic Score (Target ↑) | PickScore (Held-out ↑) | HPSv2 (Held-out ↑) |
> |--------|---:|--------:|--------------:|----------:|
> | ReNO | 0.5 | 7.1902 | 0.2142 | 0.2714 |
> | ReNO | 1.0 | 7.1931 | 0.2149 | 0.2765 |
> | ReNO | 2.0 | 6.8614 | 0.2152 | 0.2698 |
> | ReNO | 4.0 | 7.2284 | 0.2191 | 0.2887 |
> | PRNO | 0.5 | 6.9051 | 0.2109 | 0.2606 |
> | PRNO | 1.0 | 6.9507 | 0.2148 | 0.2762 |
> | PRNO | 2.0 | 7.0244 | 0.2183 | 0.2861 |
> | PRNO | 4.0 | 6.9942 | 0.2203 | 0.2979 |
> | MPGR | 0.5 | 7.1044 | 0.2146 | 0.2769 |
> | MPGR | 1.0 | 6.9868 | 0.2144 | 0.2759 |
> | MPGR | 2.0 | 6.9257 | 0.2177 | 0.2866 |
> | MPGR | 4.0 | 7.0487 | 0.2210 | 0.2934 |
> | **Ours** | **N/A** | **8.9078** | **0.2203** | **0.2986** |
>
> Baselines vary with λ (e.g., ReNO: 6.86–7.23, PRNO: 6.91–7.02, MPGR: 6.93–7.10), but even the best λ per method (ReNO: 7.23, PRNO: 7.02, MPGR: 7.10) does not approach our **8.91**. This confirms that the reported performance gap is not an artifact of the chosen λ=2.0. We will include this ablation in the revised manuscript.
>
> ---
>
> ## Limitation: Block Size Tuning
>
> **We believe the block size B is governed by Gaussian statistics rather than the task, and a single choice B=16 works across all our experiments.**
>
> Our constraints match blockwise norms to theoretical expectations. The quantities $\mathbb{E}[||\mathbf{y}^{(p)}||_1] = \sqrt{\frac{\pi}{2}} \cdot B$ and $\mathbb{E}[||\mathbf{y}^{(p)}||_2^2] = B$ are determined purely by the Gaussian distribution, independent of the reward model, generative model, or task. As shown in Table 2 (Appendix G), B ∈ {8, 16, 32} all outperform MPGR with stable held-out rewards; degradation only occurs at extremes (B=64, 32768) where constraints become too loose relative to the Gaussian distribution. In our experiments, the single choice B=16 works across all four reward models and four architectures tested, suggesting that practical tuning cost is likely minimal.

---

> > ### Author Rebuttal · Reviewer_mdgs · 2026-04-08
> >
> > Thanks for the reply. The authors have addressed all of my concerns. However, I will maintain the original evaluation.

---

### Decision · Program_Chairs · 2026-04-30

**Decision:**

Accept (regular)

**Comment:**

All reviewers agreed this paper is exceedingly well written and easy to follow, with a principled mathematical formulation and a conceptually clean, well motivated approach. The proposed PGA framework replaces soft regularization with hard white Gaussian noise constraints via an efficient closed form projection, which adds negligible overhead while effectively preventing reward hacking. It achieves comparable target rewards with significant less wall-clock time of SOTA regularization methods, with clear advantages over existing baselines.

While some reviewers’ concerns, including limited base model evaluation, lack of diversity metrics, and theoretical justification of constraint design, are reasonable in some sense, I still recommend acceptance. I encourage the authors to incorporate the suggested presentation revisions in the camera-ready version.